# Multiple contexts and frequencies aggregation network for deepfake detection

**Zifeng Li**[ID]**, Wenzhong Tang, Shijun Gao**[ID]**, Yanyang Wang, Shuai Wang***

Beihang University, Beijing, People's Republic of China

* wangshuai@buaa.edu.cn

## Abstract

Deepfake detection faces increasing challenges since the fast growth of generative models in developing massive and diverse Deepfake technologies. Recent advances rely on introducing heuristic features from spatial or frequency domains rather than modeling general forgery features within backbones. To address this issue, we turn to the backbone design with two intuitive priors from spatial and frequency detectors, i.e., learning robust spatial attributes and frequency distributions that are discriminative for real and fake samples. To this end, we propose an efficient network for face forgery detection named MkfaNet, which consists of two core modules. For spatial contexts, we design a Multi-Kernel Aggregator that adaptively selects organ features extracted by multiple convolutions for modeling subtle facial differences between real and fake faces. For the frequency components, we propose a Multi-Frequency Aggregator to process different bands of frequency components by adaptively reweighing high-frequency and low-frequency features. Comprehensive experiments on seven popular Deepfake detection benchmarks demonstrate that MkfaNet achieves an AUC of 0.9591 in within-domain evaluations and 0.7963 in cross-domain evaluations, outperforming several state-of-the-art methods while maintaining high computational efficiency. Results confirm that MkfaNet is effective and efficient in detecting forgery, offering enhanced robustness against diverse Deepfake manipulations. Our code is available at https://github.com/GGshawn/MkfaNet.

## 1 Introduction

With the development of generative models, Deepfake technology has made significant progress. Deepfake encompasses video, audio, and text, utilizing advanced artificial intelligence techniques such as Variational Autoencoders (VAE) [1], Generative Adversarial Networks (GAN) [2], and Diffusion Models (DM) [3] to achieve unprecedented realism. Unfortunately, these fake visual data can be used for malicious purposes, such as invading personal privacy, spreading misinformation, and undermining people's trust in digital media [4–6]. Considering that facial deepfakes can potentially cause more significant social and ethical implications compared to

**Data availability statement:** All relevant data are within the manuscript.

**Funding:** The author(s) received no specific funding for this work.

**Competing interests:** The authors have declared that no competing interests exist.

synthetic media without facial content, we specifically concentrate on facial deepfake technology in this paper.

To address the potential risks posed by Deepfakes, numerous researchers are working to enhance Deepfake detection technology and strengthen existing detection systems [7–13]. These methods employ various techniques and are generally classified into three types: naive detectors [14,15], spatial detectors [16,17], and frequency detectors [18,19]. Meanwhile, researchers are striving to develop sufficiently robust detectors to withstand various forms of degradation, such as noise [20–22], compression [7,23], and, most critically, to identify previously unseen Deepfakes [24, 25]. Therefore, enhancing the generalization ability of Deepfake detection models becomes particularly important. Models with strong generalization capabilities can effectively identify and counter new Deepfake attacks that have not appeared in the training data, thereby ensuring the authenticity and security of information [26].

Improving the model's ability to capture critical facial features is an effective means of enhancing its generalization capability. These key features include but are not limited to, subtle dynamics of facial expressions, natural gradients of skin tone, and natural eye blinking. By accurately capturing these difficult-to-simulate details, the model can more effectively distinguish between real content and Deepfake-generated content [21]. In recent research, adopting multitask learning [27–32] and/or heuristic fake data generation strategies [28,33] is the mainstream method to enhance the generalization capability of Deepfake detection method. These approaches aim to improve the model's adaptability and discrimination ability against novel forgery techniques by learning multiple related tasks simultaneously. Meanwhile, heuristic data generation methods create new and unseen fake samples to test and improve the robustness of detection algorithms. However, commonly used architectures for these methods, such as XceptionNet [34] and EfficientNet [35], primarily tend to learn global features while neglecting more local features [36–38]. Consequently, most of these methods fail to effectively model local artifacts, which is crucial for detecting high-quality Deepfake content.

We first focus on the differences between real and forged samples in the frequency domain, and our empirical analysis reveals significant disparities in their frequency distributions, as shown in Fig 1(b). Specifically, real samples exhibit a relatively uniform energy distribution in the spectrogram, indicating a balanced texture and edge information across various frequencies. In contrast, forged samples display abnormally concentrated energy peaks in the high-frequency region, highlighting the shortcomings of forgery techniques in handling high-frequency details, which result in unnatural textures and edges in the high-frequency area. To further illustrate this phenomenon, we use a pre-trained ResNet50 model to examine how it processes real and fake face images, with the results shown in Fig 1(c). Notably, ResNet50 exhibits a weaker response in the high-frequency region, indicating its insufficiency in capturing the high-frequency details of forged faces. Additionally, when processing forged face images, ResNet50's shallow feature maps exhibit higher low-frequency responses, whereas these low-frequency responses are weaker and more uniformly

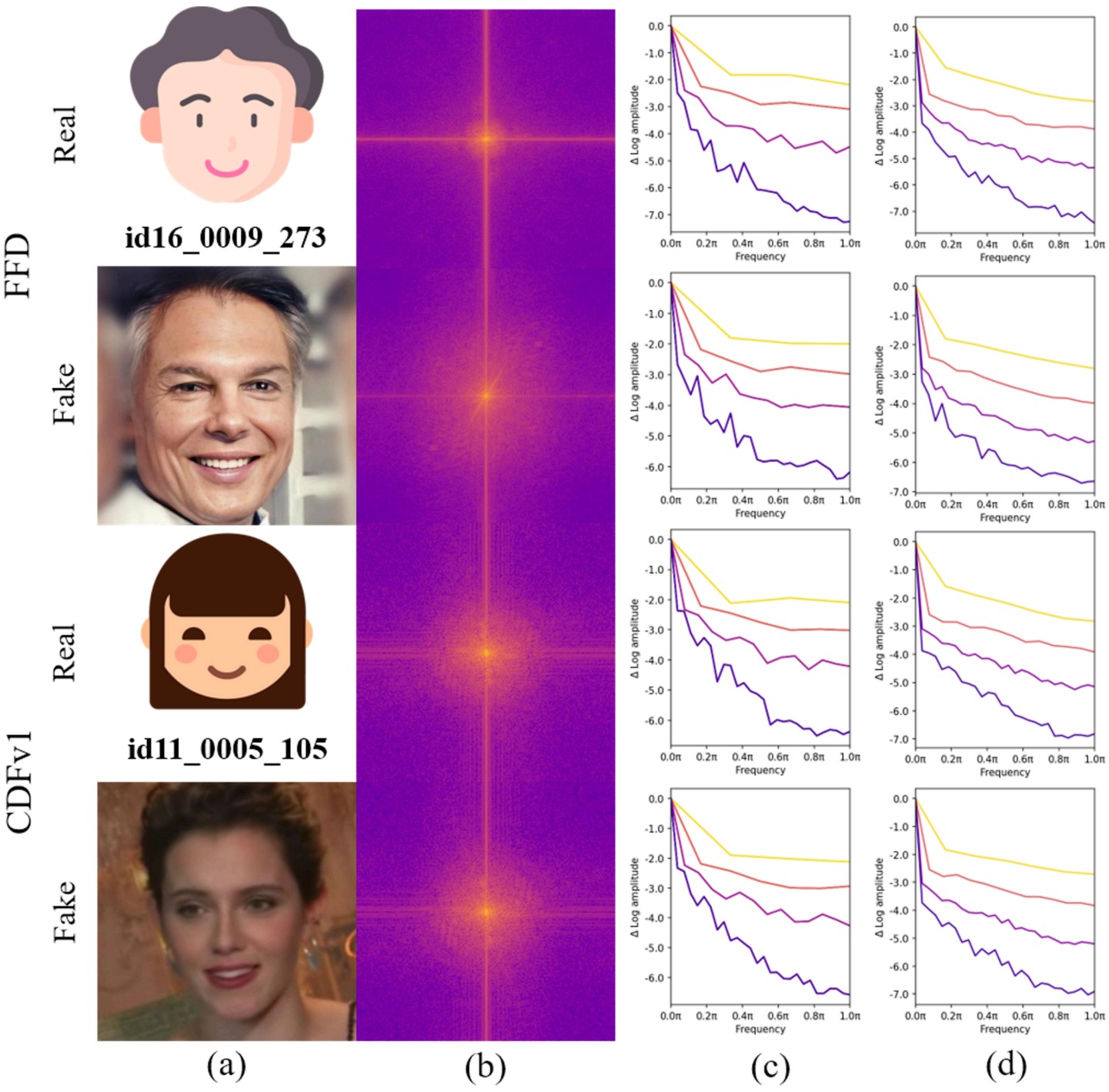

**Fig 1. Illustration of frequency priors in deepfake detection.** (a): Source image. (b): Data frequency domain analysis. (c): Relative log amplitudes of Fourier transformed feature maps of ResNet50. (d): Relative log amplitudes of Fourier transformed feature maps of MkfaNet. (b) reveals the uniformity of the frequency distribution in real faces and the concentration of high-frequency anomalies in forged faces. (c) shows that ResNet50 has a relatively low logarithmic amplitude in the high-frequency region, indicating its insufficiency in capturing high-frequency details. (d) demonstrates that MkfaNet has a higher amplitude in the high-frequency region with broader coverage, highlighting its advantages in handling high-frequency details and identifying forgery features. To protect privacy, the original facial images were anonymized by replacing them with icon representations. Corresponding image identifiers from the CelebDF–v1 dataset are shown.

distributed when processing real faces. This indicates that ResNet50 has a stronger reaction to simple features in forged images but lacks sensitivity to high-frequency details. This layered difference in frequency response reveals the underlying mechanisms by which deep networks distinguish between real and fake faces. It provides important insights and motivation for designing models that can more accurately differentiate between genuine and forged faces.

Additionally, we have observed that recent advances rely on introducing heuristic features from either the spatial or frequency domain, rather than establishing a general forgery feature detection model within the backbone network. While this approach improves detection performance to some extent, it still has limitations, especially in addressing the continuously evolving forgery techniques. Therefore, we propose MkfaNet, which integrates more powerful feature capture and analysis capabilities into the backbone network by combining the Multi-Kernel Aggregator (MKA) and Multi-Frequency Aggregator (MFA), significantly enhancing the accuracy and robustness of forgery detection. In specific, the Multi-Kernel Aggregator (MKA) module combines depth-wise separable convolutions with different dilation rates to effectively expand the model's receptive field, enhancing its ability to capture features at various scales from the input data. It then adaptively selects features extracted through multiple convolutions based on the spatial context to model the subtle facial differences between real and fake faces; Multi-Frequency Aggregator (MFA) module optimizes the model's response to different frequency information by separately processing and fusing the DC (Direct Current) and HC (High Current) components of images. MkfaNet, as a stack of MKA and MFA modules, shows the enhanced ability to discern image details and structural information. In the context of real and fake face recognition, it can accurately distinguish the subtle texture and frequency distortions introduced by forgery techniques, thereby improving the accuracy of fake image detection.

Deepfake techniques such as face swapping and facial expression modification pose unique detection challenges. Face swapping often introduces blending artifacts and identity inconsistencies, making it difficult to detect without spatial-aware feature extraction. Similarly, facial expression modification creates subtle yet unnatural deformations, which are better captured through frequency-based analysis. These challenges highlight the need for a detection framework that can effectively model both spatial and frequency discrepancies, motivating the design of MkfaNet.

Comprehensive experiments on seven popular deepfake detection benchmarks [39] demonstrate that our proposed MkfaNet variants achieve superior performances in both within-domain and across-domain evaluations with impressive efficiency of parameter usage.

This work mainly makes the following contributions:

- In deepfake detection, accurately capturing multi-scale details is crucial because forgery techniques often intervene and modify image features at various scales. The MKA module enhances the model's receptive field with convolutional kernels of different dilation rates, allowing the model to more effectively capture features from fine textures to larger structures, thus improving the ability to recognize subtle signs of forgery.
- Deepfake technology often manipulates high-frequency details to achieve face swapping or modification, which may result in unnatural features across different frequencies. The MFA module optimizes the model's response to these critical frequency information by independently processing and integrating the Direct Current (DC) and High Current (HC) frequency components, significantly enhancing the sensitivity and recognition capability for high-frequency detail anomalies.
- Extensive testing of MkfaNet on various mainstream deepfake detection benchmarks demonstrates its effectiveness and efficiency in handling different deepfake techniques. This also shows MkfaNet's advantage in parameter efficiency, which is extremely important for practical applications.

To provide a structured discussion, the rest of this paper is organized as follows. Sect 2 reviews related work, summarizing existing deepfake detection approaches and their limitations. Sect 3 introduces our proposed MkfaNet, detailing its architecture and the design of the Multi-Kernel Aggregator (MKA) and Multi-Frequency Aggregator (MFA) modules. Sect 4 presents the experimental setup and results, including dataset descriptions, evaluation metrics, and comparative analysis

with state-of-the-art methods. Finally, Sect 5 concludes the paper, summarizing key findings and outlining potential future research directions.

## 2 Related work

**Deepfake generation.** Deepfake technology primarily involves the artificial modification of facial images and has significantly evolved since its inception. Since 2017, machine learning-based facial manipulation techniques have made substantial advancements, particularly in the areas of facial replacement and facial expression reenactment, which have garnered widespread attention [39]. Ian Goodfellow et al. introduced Generative Adversarial Networks (GANs) [40], a technology that has significantly advanced the development of realistic image synthesis, including facial images [41,42]. GANs consist of two parts: the generator and the discriminator. The generator is responsible for creating images, while the discriminator's task is to distinguish between these generated images and real data. Variational Autoencoders (VAEs) [43] compress data into a compact form and are used in Deepfake technology to alter facial features, such as expressions and styles. Diffusion models (DMs) [44,45] create images by gradually adding noise and then progressively removing this noise during the generation process. In facial image generation, diffusion models can produce high-quality, high-resolution facial images by finely controlling the noise reduction process. Facial Deepfakes can be broadly categorized into two types: face-swapping and face-reenactment. Face-swapping refers to replacing the facial features in one image with the facial features from another image [46–48]. Face-reenactment technology modifies the original face using image processing techniques to mimic the expressions of another face. Face2Face [49] generates different expressions by tracking facial key points, while NeuralTextures [50] achieves expression transfer using rendered images generated from 3D facial models. These technologies enable more diverse and precise simulation of facial expressions.

**Deepfake detection.** In Deepfake detection research, methods can be broadly categorized into image-level detectors and video-level detectors. Image-level detectors analyze individual frames to identify fake images by recognizing spatial artifacts. One widely used approach is the Xception model [15], a convolutional neural network (CNN) architecture often combined with attention mechanisms like MAT [37] to enhance feature extraction. Other methods, such as Face X-ray [28], leverage the boundaries between forged faces and backgrounds to detect spatial inconsistencies. More recently, algorithms have focused on detecting blending artifacts [9,51] or learning to separate relevant and irrelevant features during training, aiming to improve generalization. Additionally, feature selection and semi-supervised learning techniques [31, 32] have been explored to enhance robustness by improving feature representation and leveraging unlabeled data. Video-level detectors, in contrast, utilize temporal information from multiple frames to enhance deepfake video detection [52]. For example, FTCN [53] directly extracts temporal information using 3D CNNs with a spatial kernel size of 1, while Alt-Freeze [54] improves generalization by independently training spatial and temporal features. Despite their promising performance, existing deepfake detection methods face several critical challenges:

**Over-reliance on heuristic features**: Many models rely heavily on manually designed spatial or frequency features, which may not generalize well across different datasets and deepfake techniques.

**Limited backbone architectures**: The majority of approaches use traditional DNN backbones such as Xception-Net [34] and EfficientNet [35], which inherently extract global features through deep convolutional layers. This architecture design can lead to the loss of critical localized forgery artifacts, thereby reducing detection robustness.

**Challenges in detecting high-quality deepfakes**: As deepfake technology advances, newer forgery techniques produce more realistic facial textures and seamless blending, making them increasingly difficult to detect with existing methods. Many models struggle with capturing subtle manipulation traces, especially in high-resolution deepfakes.

These challenges highlight the necessity of designing a more effective detection backbone that can adaptively extract discriminative spatial and frequency features while maintaining strong generalization. Our proposed MkfaNet addresses these limitations by incorporating Multi-Kernel Aggregation (MKA) and Multi-Frequency Aggregation (MFA) modules,

which improve the ability to capture both spatial and frequency-based artifacts, enhancing the model's robustness against sophisticated forgery techniques.

## 3 Method

### 3.1 Overview of MkfaNet

Built upon modern ConvNets, we design a four-stage MkfaNet architecture as illustrated in Fig 2. The overall detection pipeline and architectural design of MkfaNet are illustrated in Fig 2, which outlines the flow from input image through hierarchical feature extraction modules to the final classification output. For stage $i$, the input image or feature is first fed into an embedding stem to regulate the resolutions and embed into $C_i$ dimensions. Assuming the input image in $H \times W$ resolutions, features of the four stages are in $\frac{H}{4} \times \frac{W}{4}$, $\frac{H}{8} \times \frac{W}{8}$, $\frac{H}{16} \times \frac{W}{16}$, and $\frac{H}{32} \times \frac{W}{32}$ resolutions respectively. Then, the embedded feature flows into $N_i$ Mkfa Blocks, consisting of spatial and channel aggregation blocks, for multi-kernel feature and high-low frequency aggregation.

### 3.2 Multi-kernel aggregator

Model generative models are able to create extremely realistic fake human faces that are visually almost indistinguishable from real ones by learning from a vast amount of real facial data and thus simulating features such as lighting, texture, and shape of faces. In this context, traditional single-scale feature extraction methods struggle to detect these fake faces. This is mainly because such methods typically focus on features at a fixed scale, such as coarse patterns of edges or textures, and overlook the subtle changes and complex interactions across multiple scales, which are precisely what generation techniques excel at simulating.

Therefore, to effectively distinguish these high-quality fake images from real faces, a method capable of analyzing and identifying details at multiple levels is required. In this way, we propose MKA modules as a solution, which adaptively selects organ features extracted by multiple convolutions for modeling subtle facial differences between real and fake faces. To elucidate the implementation details of the Multi-kernel Aggregator (MKA) module, as illustrated in Fig 3(a), we will delve into its architectural design, focusing on how it adaptively aggregates multi-level features to enhance the detection of key facial regions. We represent this process as follows:

$$Z = X + \text{MKA}(\text{Norm}(X)), \tag{1}$$

where MKA($\cdot$) denotes a multi-kernel gated aggregation module comprising the gating $\mathcal{F}_\phi(\cdot)$ and multi-kernel feature branch $\mathcal{G}_\psi(\cdot)$.

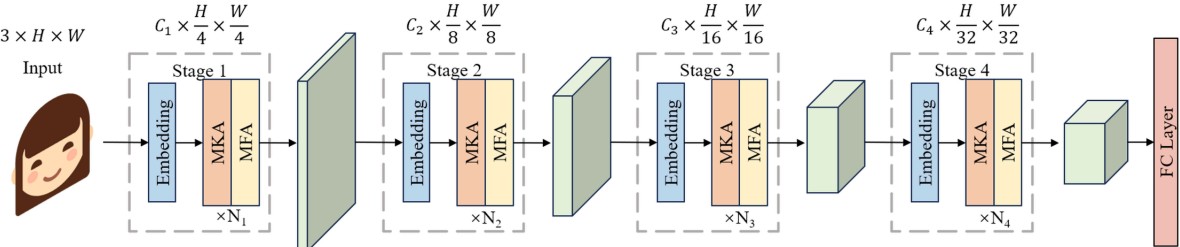

**Fig 2. Overall detection pipeline of MkfaNet.** The model consists of a four-stage hierarchical backbone, where each stage includes an embedding stem, Multi-Kernel Aggregator (MKA), and Multi-Frequency Aggregator (MFA). The pipeline takes an input face image and processes it through the stacked modules to produce a final classification score via a fully connected (FC) layer.

**Multi-kernel feature extraction.** To enable the model to perceive the multi-level features of the face images, we employ three different DWConv layers with dilation ratios $d \in \{1, 2, 3\}$ in parallel to capture low, middle, and high-order features: given the input feature $X \in \mathbb{R}^{C \times HW}$, the input is factorized into $X_l \in \mathbb{R}^{C_l \times HW}$, $X_m \in \mathbb{R}^{C_m \times HW}$, and $X_h \in \mathbb{R}^{C_h \times HW}$ along the channel dimension, where $C_l + C_m + C_h = C$; afterward, $X_l$, $X_m$ and $X_h$ are assigned to $\text{DW}_{7 \times 7, d=1}$, $\text{DW}_{7 \times 7, d=2}$ and $\text{DW}_{7 \times 7, d=3}$, respectively. Finally, the output of $X_l$, $X_m$, and $X_h$ are concatenated to form multi-kernel feature, so that $Y_C = \text{Concat}(Y_l, Y_m, Y_h)$.

**Gated aggregation.** To *adaptively* aggregate the extracted feature from the multi-kernel feature branch, and we employ SiLU activation in the gating branch, as $\text{SiLU}(x) = x \cdot \text{Sigmoid}(x)$, which has been well-acknowledged as an advanced version of Sigmoid activation. SiLU has both the gating effect of Sigmoid and stable training characteristics, leading the final aggregated features as

$$Z = \underbrace{\text{SiLU}(\text{Conv}_{1 \times 1}(X))}_{\mathcal{F}_\phi} \odot \underbrace{\text{SiLU}(\text{Conv}_{1 \times 1}(Y_C))}_{\mathcal{G}_\psi}. \tag{2}$$

### 3.3 Multi-frequency aggregator

Fig 1(b) shows the frequency domain analysis of the data, revealing significant differences in the distribution of high-frequency information between fake and real faces. Fake faces often appear unnatural in details such as skin texture and edge sharpness, resulting in a noticeably different distribution of features in the high-frequency region. Fig 1(c) illustrates the relative logarithmic amplitude of the Fourier-transformed data, with the color gradient from purple to yellow representing the transition from shallow to deep layers of the model. This gradient reveals how layers of different depths handle frequency information, providing visual evidence of the differences in frequency responses between real and fake faces.

It is evident that the shallow layers (purple) tend to capture high-frequency details related to texture and edges, while the deeper layers (yellow) strongly respond to low-frequency features, which are typically associated with the overall structure and shape of the image. At these levels, real and fake faces exhibit different frequency characteristics. Specifically, in the high-frequency details, fake faces often fail to perfectly replicate the high-frequency features of real faces due to technical limitations, resulting in anomalies or inconsistencies in the high-frequency region. This underscores the importance of addressing both low-frequency and high-frequency features in facial recognition.

We propose an MFA module that processes and reorganizes the direct current (DC) and high-frequency (HC) components of images independently, allowing the model to perform more refined and in-depth analysis at different frequency levels. Specifically, the MFA enhances the analysis of high-frequency details to identify unnatural textures and edges produced by generative models while integrating low-frequency information to maintain an understanding of the overall structure of the image. This approach not only strengthens the model's ability to detect flaws unique to forgery techniques but also improves its capacity to capture authentic features. As a result, the accuracy and robustness of facial authenticity recognition are significantly enhanced. By comprehensively analyzing features at different frequencies, the MFA helps the model better distinguish and recognize complex real and fake faces, effectively addressing the challenges posed by high-quality forgery techniques. The structure of the MFA module, as shown in Fig 3(b), can be formalized as follows:

$$Y = \text{GELU}\big(\text{DW}_{3 \times 3}\big(\text{Conv}_{1 \times 1}(\text{Norm}(X))\big)\big),$$
$$Z = \text{Conv}_{1 \times 1}\big(\text{MF}(Y)\big) + X. \tag{3}$$

where $\text{MF}(\cdot)$ is a scaling technique that operates on feature maps by distinctively mixing information from different frequency bands. In specific, the input signal is firstly decomposed into its DC component and high-frequency components. Then, two sets of parameters are introduced to re-weight these components for each channel. The two-step processing

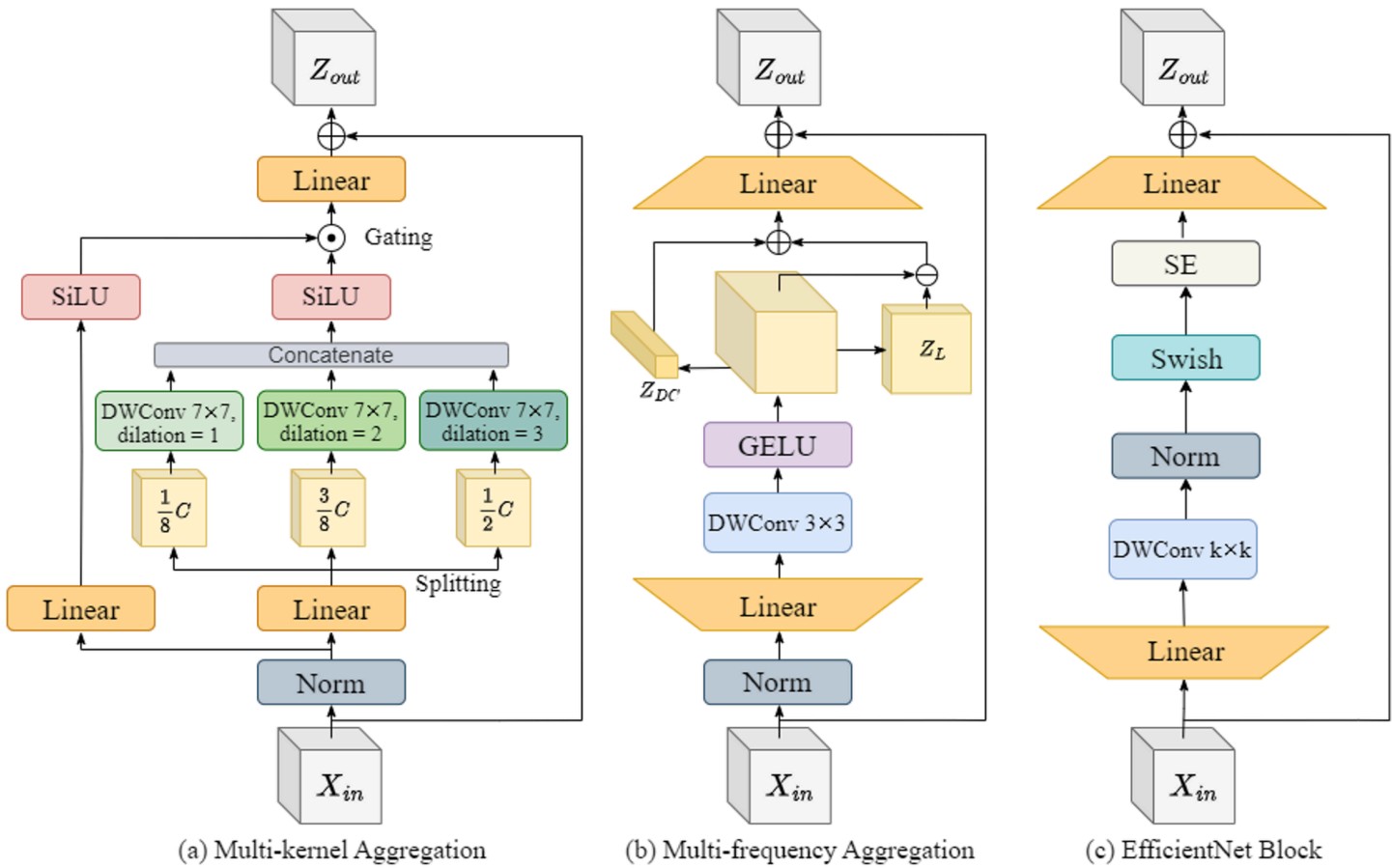

**Fig 3. (a) Multi-Kernel Aggregation (MKA) block, designed as a token mixer, utilizes depthwise convolution layers with different dilation rates to capture multi-scale spatial features, improving sensitivity to subtle local manipulations.** (b) Multi-Frequency Aggregation (MFA) block, serving as a channel mixer, applies frequency-aware feature decomposition through depthwise convolutions and gating mechanisms to enhance forgery-specific artifact extraction. (c) EfficientNet block, which integrates a Squeeze-and-Excitation (SE) module to adaptively recalibrate feature responses. (d) ConvNeXt block, incorporating a Channel Mixer and depthwise convolutions, serves as a baseline for comparing hierarchical feature mixing strategies.

reads as,

$$Y_{DC} = z_{DC} \odot Y,$$
$$Y_{HC} = Y - z_L \odot Y, \tag{4}$$
$$MF(Y) = Y_{DC} + \gamma \odot Y_{HC},$$

where $\gamma$ is the channel-wise scaling factor initialized as zeros. To ensure the efficient computation of high- and low-frequency components, we referred to the method of Wang et al. [55], rather than using an explicit Fourier transform. The DC component is calculated by averaging each feature map, while the HC component is obtained by subtracting the DC component from the original features. Specifically, $z_{DC}$ represents the spatial average, and $z_L$ represents the channel average.

## 3.4 Discussion

To further highlight the unique design of our proposed MkfaNet, we compare it with several representative backbones in terms of their feature extraction strategies and the presence of spatial and frequency aggregation mechanisms, as shown

in Table 1. Unlike existing models such as XceptionNet, EfficientNet, and ConvNeXt, which either lack frequency modeling or rely solely on global features, MkfaNet explicitly incorporates both multi-scale spatial feature aggregation and frequency-aware processing. This dual-branch architecture enables MkfaNet to more effectively capture subtle local artifacts and abnormal frequency patterns, thereby enhancing its robustness against high-quality and cross-domain Deepfakes.

**3.4.1 Advantages over classical CNN.** Currently, the most commonly used architectures for fake face detection are XceptionNet [34] and EfficientNet [35]. XceptionNet builds its structure using depthwise separable convolution layers(DWConv), optimizing the learning of global features [36,37], which makes the model excel in recognizing overall image structures and patterns. However, for tasks that require detailed analysis of local features, such as fake face detection, this approach may limit the model's sensitivity to subtle facial expression differences and skin texture patterns. EfficientNet (Fig 3(c)), on the other hand, enhances model efficiency and feature representation capability by balancing network depth, width, and resolution adjustments, combined with depthwise separable convolutions and Squeeze-and-Excitation (SE) blocks. Although SE blocks improve the model's attention to features, this global information-based recalibration fails to capture the local detail anomalies unique to forged faces.

Both XceptionNet and EfficientNet employ depthwise separable convolutions to simultaneously process spatial and channel features, limiting their ability to perceive specific contextual with tiny differences and channel features of different frequencies. Our proposed MkfaNet's superiority over them for fake face detection lies in its combination of two core modules, which respectively focus on learning spatial context and channel features, which expand the receptive field by using depth-wise separable convolutions with different dilation rates and excels in detecting subtle anomalies introduced by strengthening the extraction of high-frequency details. In this way, in comparison to the XceptionNet and EfficientNet, MkfaNet can learn richer features, providing stronger identification capabilities and higher reliability when dealing with complex facial data and advanced forgery techniques.

**3.4.2 Advantages over modern DNN.** By employing block-based designs combined with hierarchical and isotropic stages, modern DNNs can effectively handle large-scale and complex datasets, capture long-range dependencies, and perform multi-scale feature extraction. Additionally, these networks can adaptively adjust the functionality and dimensions of each layer, providing greater flexibility to meet different task requirements, thereby significantly improving model performance while maintaining parameter efficiency. ConvNext [56] (Fig 3(b)) as a modern convolutional neural network architecture, separates the processing of spatial features and channel features and uses an additional channel mixer to enhance the interaction between different channels, thereby enriching feature representation. However, ConvNext uses only one depthwise separable convolution for spatial features, and the channel mixer simply performs an up-and-down dimensional operation on the channels. While these designs improve inter-channel interaction, their sensitivity to local detail features may still be insufficient. Therefore, although this model performs well in general image tasks, it may require further adjustments or integration with other mechanisms in specialized fake face detection tasks to capture better and analyze the inherent local and high-frequency detail features of forgery techniques.

**Table 1. Comparison of models in feature extraction and aggregation strategies.**

| Method | Extraction | Spatial Agg | Freq Agg |
|---|---|---|---|
| XceptionNet | DWS Conv | No | No |
| EfficientNet | DWS Conv | No | No |
| | SE | No | No |
| ConvNeXt | DW conv | Yes | No |
| | Channel mixer | Yes | No |
| **MkfaNet (Ours)** | MKA | Yes | No |
| | MFA | No | Yes |

On this basis, considering the special requirements of deepfake detection, our MkfaNet model offers significant advantages: Built upon a modern DNN architecture, MkfaNet enhances the ability to capture local and high-frequency details in deepfake images by integrating MKA and MFA. The MKA module adaptively selects features of specific organs through multiple convolution processes, accurately simulating the subtle differences between real and fake faces, while the MFA module focuses on frequency components, adaptively rebalancing high- and low-frequency features to increase the model's sensitivity to abnormal high-frequency details in fake images. This gives MkfaNet higher accuracy and efficiency compared to traditional, modern DNN architectures, especially demonstrating superior performance in handling advanced forgery techniques.

### 3.5 Network configurations

We provide detailed architecture configurations of MkfaNet variants in Table 2, where we scale the embedding dimensions and the number of blocks for each stage. (1) MkfaNet-Tiny, with embedding dimensions of 32,64,128,256, is designed for lightweight deepfake detection scenarios and contains only 5.2M parameters and 1.5 GFLOPs. It achieves a fast inference time of 22 ms on a batch of 480 images tested on an RTX 4090 GPU. (2) MkfaNet-Small uses larger embedding dimensions 64,128,320,512, reaching 19.8M parameters and 6.7 GFLOPs, while still maintaining a practical inference time of 98 ms under the same setting. Compared with other modern architectures such as ConvNeXt [56], which typically involve around 25M parameters, MkfaNet-Small provides a favorable trade-off between performance and computational efficiency.

## 4 Experiments

### 4.1 Settings

**Datasets.** To evaluate the performances and generalization abilities of our proposed backbone, we follow DeepfakeBench [39] to conduct comparison and analysis experiments on seven commonly used deepfake detection datasets, as shown

**Table 2**. Architecture configurations of MkfaNet variants.

| Stage | Output Size | Layer Settings | MkfaNet | |
|---|---|---|---|---|
| | | | Tiny | Small |
| S1 | $\frac{H \times W}{4 \times 4}$ | Stem | $Conv_{3 \times 3}$, stride 2, $C/2$ $Conv_{3 \times 3}$, stride 2, $C$ | |
| | | Embed. Dim. | 32 | 64 |
| | | # Mkfa Block | 3 | 2 |
| | | MLP Ratio | 8 | |
| S2 | $\frac{H \times W}{8 \times 8}$ | Stem | $Conv_{3 \times 3}$, stride 2 | |
| | | Embed. Dim. | 64 | 128 |
| | | # Mkfa Block | 3 | 3 |
| | | MLP Ratio | 8 | |
| S3 | $\frac{H \times W}{16 \times 16}$ | Stem | $Conv_{3 \times 3}$, stride 2 | |
| | | Embed. Dim. | 128 | 320 |
| | | # Mkfa Block | 12 | 10 |
| | | MLP Ratio | 4 | |
| S4 | $\frac{H \times W}{32 \times 32}$ | Stem | $Conv_{3 \times 3}$, stride 2 | |
| | | Embed. Dim. | 256 | 512 |
| | | # Mkfa Block | 2 | 2 |
| | | MLP Ratio | 4 | |
| Parameters (M) | | | 5.2 | 19.8 |
| FLOPs (G) | | | 1.5 | 6.7 |
| Inference times (ms) | | | 22 | 98 |

in Table 3: FaceForensics++ (FF++) [15], CelebDF-v1 (CDFv1) [65], CelebDF-v2 (CDFv2) [65], DeepFakeDetection (DFD) [66], DeepFake Detection Challenge Preview (DFDC-P) [67], DeepFake Detection Challenge (DFDC) [68], and DeeperForensics-1.0 (DF-1.0) [20]. Specifically, FF++ is a large-scale database with 1.8 million forged images that contains 4 types of manipulation methods, including Deepfakes (FF-DF) [69], Face2Face (FF-F2F) [49], FaceSwap (FF-FS) [70], and NeuralTextures (FF-NT) [50]. Note that we use the lightly compressed (c23) version of FF++ as the default training data, whereas two other compressed versions of FF++ are raw and heavily compressed (c40), while others are used as testing datasets.

Beyond FF++, CelebDF-v1 and CelebDF-v2 introduce more challenging Deepfake synthesis techniques that mitigate the typical visual artifacts found in FF++, making them closer to real-world Deepfake videos. CelebDF-v2 further refines CelebDF-v1 by reducing boundary artifacts and unnatural lip movements, increasing the difficulty of detection. DFD is a high-quality deepfake dataset released by Google, containing videos forged using multiple face-swapping techniques, captured in a controlled environment with consistent lighting. DFDC-P and DFDC were developed by Facebook and include a diverse set of Deepfake videos created with multiple face manipulation techniques, representing real-world variations in lighting conditions, compression levels, and face occlusions. DFDC is especially challenging due to its mix of synthetic and real data, as well as multiple unknown manipulation techniques. DF-1.0 is designed to evaluate model robustness under various real-world perturbations, such as Gaussian noise, compression artifacts, color distortions, and adversarial attacks. These perturbations simulate realistic degradation scenarios, making DF-1.0 highly suitable for assessing cross-domain generalization capabilities.

To ensure fair and consistent evaluations, We adopt the full data pre-processing workflow proposed in DeepfakeBench and use the fixed training and testing resolutions of $256 \times 256$ for the cropped face images.

**Implementation details.** For a fair comparison, we consider three types of detectors in DeepfakeBench [39], as detailed in Table 4: **(1) Naive detectors** that combine a backbone and binary classifier without introducing manually designed features. Both classical CNNs (e.g., ResNet [71] and EfficientNet [35]) and modern architectures (e.g., Swin Transformer [59] and ConvNeXt [56]) are compared. **(2) Spatial detectors** that build upon the backbone and further utilize spatial features with manually designed algorithms. **(3) Frequency detectors** focus on exploring frequency components and artifacts to detect forgeries. As for training settings, we follow the official data splits provided by DeepfakeBench [39] to ensure fair and consistent evaluation. All detectors with classical CNN backbones are trained using the Adam optimizer with a learning rate of $2 \times 10^{-4}$ and a batch size of 32. For detectors based on modern backbones, such as ConvNeXt and our MkfaNet, we adopt the AdamW optimizer with a learning rate of $5 \times 10^{-4}$ and a batch size of 256. Pre-trained weights on ImageNet-1K are used for backbone initialization when available, and MkfaNet adopts the same pre-training setting as ConvNeXt-T, as shown in Table 5. We apply several data augmentations, including image compression, horizontal flipping, rotation, Gaussian blur, and random brightness/contrast adjustment. All models are trained for 50 epochs, and the best-performing checkpoint is selected based on validation performance. For evaluation, we report the mean frame-level Area Under the Curve (AUC) over three trials.

## 4.2 Comparison results

As shown in Table 4, we conduct within-domain and cross-domain evaluations for three versions of our MkfaNet, i.e., the lightweight detectors, the naive detectors compared to various backbones, and the advanced detectors with prior knowledge from spatial or frequency domains. cross-domain evaluation involves testing the model on different datasets.

**Within-domain evaluations.** We first conduct within-domain evaluations to verify the performances of detectors within the same dataset following DeepfakeBench [39]. Table 4 (middle columns) shows MkfaNet variants achieve the best average results on six within-domain datasets. As for the lightweight detectors, MkfaNet-T significantly outperforms MesoNet [14] and CapsuleNet [57] with similar parameters by 9.33% and 2.87% AUC while outperforming CNN-Aug [58]

**Table 3. Within-domain and cross-domain evaluations of various deepfake detectors and backbones using the AUC metric.** All detectors are trained on FF-c23 and eval-uated on other datasets. **Avg.** donates the average AUC for within-domain and cross-domain evaluations, and the best result for each group is highlighted in **bord.** † represents our reproduced results, while DeepfakeBench provides others. The values reported are accompanied by ±, which represents the margin of error corresponding to the values in the previous row, and the confidence interval is at a 95% confidence level.

| Type | Detector | Backbone | # Param. (M) | Within Domain Evaluation | | | | | | | Cross Domain Evaluation | | | | | | |
|---|---|---|---|---|---|---|---|---|---|---|---|---|---|---|---|---|---|
| | | | | FF-c23 | FF-c40 | FF-DF | FF-F2F | FF-FS | FF-NT | Avg. | CDFv1 | CDFv2 | DF-1.0 | DFD | DFDC | DFDCP | Avg. |
| Naive | Meso4 [14] | MesoNet | 0.03 | 0.6077 | 0.5920 | 0.6771 | 0.6170 | 0.5946 | 0.5701 | 0.6097 | 0.7358 | 0.6091 | 0.9113 | 0.5481 | 0.5560 | 0.5994 | 0.6599 |
| Naive | MesoIncep [14] | MesoNet | 0.03 | 0.7583 | 0.7278 | 0.8542 | 0.8087 | 0.7421 | 0.6517 | 0.7571 | 0.7366 | 0.6966 | 0.9233 | 0.6069 | 0.6226 | **0.7561** | 0.7237 |
| Spatial | Capsule [57] | Capsule | 4.0 | 0.8421 | 0.7040 | 0.8669 | 0.8634 | 0.8734 | **0.7804** | 0.8217 | **0.7909** | 0.7472 | 0.9107 | 0.6841 | 0.6465 | 0.6568 | 0.7394 |
| Naive | CNN-Aug [58] | ResNet-34 | 22 | 0.8493 | 0.7846 | **0.9048** | 0.8788 | 0.9026 | 0.7313 | 0.8419 | 0.7420 | 0.7027 | 0.7993 | 0.6464 | 0.6361 | 0.6170 | 0.6906 |
| Naive | **MkfaNet** | **MkfaNet-T** | 5.2 | **0.8506** | **0.7879** | 0.8982 | **0.8823** | **0.9037** | 0.7796 | **0.8504** | 0.7881 | **0.7486** | **0.9245** | **0.6883** | **0.6477** | 0.7428 | **0.7567** |
| | | | | ±0.0004 | ±0.0007 | ±0.0011 | ±0.0012 | ±0.0010 | ±0.0008 | | ±0.0002 | ±0.0004 | ±0.0005 | ±0.0007 | ±0.0019 | ±0.0011 | |
| Naive | CNN-Aug [58] | ResNet-50† | 26 | 0.8925 | 0.7956 | 0.9258 | 0.9135 | 0.9252 | 0.7828 | 0.8726 | 0.7608 | 0.7591 | 0.8143 | 0.6995 | 0.6661 | 0.6245 | 0.7207 |
| Naive | Xception [15] | Xception | 23 | 0.9637 | 0.8261 | 0.9799 | 0.9785 | 0.9833 | 0.9385 | 0.9450 | 0.7794 | 0.7365 | 0.8341 | 0.8163 | 0.7077 | 0.7374 | 0.7686 |
| Naive | Efficient [35] | Efficient-B4 | 19 | 0.9567 | 0.8150 | 0.9757 | 0.9758 | 0.9797 | 0.9308 | 0.9389 | 0.7909 | 0.7487 | 0.8330 | 0.8148 | 0.6955 | 0.7283 | 0.7685 |
| Naive | Swin [59] | Swin-T† | 28 | 0.9630 | 0.8278 | 0.9802 | 0.9783 | 0.9826 | 0.9375 | 0.9449 | 0.7863 | 0.7476 | 0.8384 | 0.8028 | 0.7053 | 0.7339 | 0.7691 |
| Naive | ConvNeXt [56] | ConvNeXt-T† | 29 | 0.9644 | 0.8287 | 0.9796 | 0.9801 | 0.9840 | 0.9393 | 0.9460 | 0.7837 | 0.7491 | 0.8425 | 0.8102 | 0.7075 | 0.7366 | 0.7691 |
| Naive | **MkfaNet** | **MkfaNet-S** | 20 | **0.9671** | **0.8315** | **0.9826** | **0.9820** | **0.9849** | **0.9428** | **0.9485** | **0.7946** | **0.7538** | **0.8785** | **0.8166** | **0.7127** | **0.7413** | **0.7824** |
| | | | | ±0.0013 | ±0.0010 | ±0.0010 | ±0.0005 | ±0.0008 | ±0.0015 | | ±0.0011 | ±0.0009 | ±0.0018 | ±0.0009 | ±0.0014 | ±0.0016 | |
| Frequency | F3Net [19] | Xception | 23 | 0.9635 | 0.8271 | 0.9793 | 0.9796 | 0.9844 | 0.9354 | 0.9449 | 0.7769 | 0.7352 | 0.8431 | 0.7975 | 0.7021 | 0.7354 | 0.7650 |
| Frequency | SPSL [60] | Xception | 23 | 0.9610 | 0.8174 | 0.9781 | 0.9754 | 0.9829 | 0.9299 | 0.9408 | **0.8150** | 0.7650 | **0.8767** | 0.8122 | 0.7040 | 0.7408 | 0.7856 |
| Frequency | SRM [18] | Xception | 23 | 0.9576 | 0.8114 | 0.9733 | 0.9696 | 0.9740 | 0.9295 | 0.9359 | 0.7926 | 0.7552 | 0.8638 | 0.8120 | 0.6995 | 0.7409 | 0.7773 |
| Frequency | F3Net [19] | ConvNeXt-T† | 29 | 0.9670 | 0.8282 | 0.9805 | 0.9858 | 0.9869 | 0.9401 | 0.9481 | 0.7753 | 0.7394 | 0.8495 | 0.8023 | 0.7106 | 0.7376 | 0.7739 |
| Frequency | F3Net [19] | **MkfaNet-S** | 20 | **0.9703** | **0.8315** | **0.9816** | **0.9867** | **0.9901** | **0.9428** | **0.9505** | **0.7957** | **0.7658** | 0.8623 | **0.8154** | **0.7136** | **0.7415** | **0.7860** |
| | | | | ±0.0011 | ±0.0009 | ±0.0014 | ±0.0013 | ±0.0012 | ±0.0010 | | ±0.0016 | ±0.0007 | ±0.0019 | ±0.0008 | ±0.0012 | ±0.0006 | |
| Spatial | FWA [61] | Xception | 23 | 0.8765 | 0.7357 | 0.9210 | 0.9000 | 0.8843 | 0.8120 | 0.8549 | 0.7897 | 0.6680 | 0.9334 | 0.7403 | 0.6132 | 0.6375 | 0.7303 |
| Spatial | X-ray [28] | HRNet | 22 | 0.9592 | 0.7925 | 0.9794 | **0.9872** | 0.9871 | 0.9290 | 0.9391 | 0.7093 | 0.6786 | 0.5531 | 0.7655 | 0.6326 | 0.6942 | 0.6722 |
| Spatial | FFD [62] | Xception | 22 | 0.9624 | 0.8237 | 0.9803 | 0.9784 | 0.9853 | 0.9306 | 0.9434 | 0.7840 | 0.7435 | 0.8609 | 0.8024 | 0.7029 | 0.7426 | 0.7727 |
| Spatial | CORE [63] | Xception | 22 | 0.9638 | 0.8194 | 0.9787 | 0.9803 | 0.9823 | 0.9339 | 0.9431 | 0.7798 | 0.7428 | 0.8475 | 0.8018 | 0.7049 | 0.7341 | 0.7685 |
| Spatial | UCF [64] | Xception | 47 | 0.9705 | 0.8399 | 0.9883 | 0.9840 | 0.9896 | 0.9441 | 0.9527 | 0.7793 | 0.7527 | 0.8241 | 0.8074 | 0.7191 | 0.7594 | 0.7737 |
| Spatial | FFD [62] | **MkfaNet-S** | 20 | **0.9829** | **0.8475** | **0.9916** | 0.9869 | **0.9937** | **0.9524** | **0.9591** | **0.8065** | **0.7679** | 0.8930 | **0.8194** | **0.7258** | **0.7652** | **0.7963** |
| | | | | ±0.0021 | ±0.0019 | ±0.0020 | ±0.0017 | ±0.0016 | ±0.0011 | | ±0.0012 | ±0.0012 | ±0.0023 | ±0.0014 | ±0.0017 | ±0.0009 | |

**Table 4**. Summary of used deepfake detection datasets.

| Dataset | Domain | Real Videos | Fake Videos | Total Videos | Rights Cleared | Total Subjects | Synthesis Methods |
|---|---|---|---|---|---|---|---|
| FF++ [15] | Within | 1000 | 4000 | 5000 | NO | N/A | 4 |
| FaceShifter [46] | Cross | 1000 | 1000 | 2000 | NO | N/A | 1 |
| DFD [66] | Cross | 363 | 3000 | 3363 | YES | 28 | 5 |
| DFDC-P [67] | Cross | 1131 | 4119 | 5250 | YES | 66 | 2 |
| DFDC [68] | Cross | 23,654 | 104,500 | 128,154 | YES | 960 | 8 |
| CelebDF-v1 [65] | Cross | 408 | 795 | 1203 | NO | N/A | 1 |
| CelebDF-v2 [65] | Cross | 590 | 5639 | 6229 | NO | 59 | 1 |
| DF-1.0 [20] | Cross | 50,000 | 10,000 | 60,000 | YES | 100 | 1 |

**Table 5**. Hyper-parameters and training recipes for ImageNet-1K of Swin-T, ConvNeXt-T, and our proposed MkfaNet-T/S.

| Configuration | Swin Tiny | ConvNeXt Tiny | MkfaNet Tiny/Small |
|---|---|---|---|
| Input resolution | $224^2$ | $224^2$ | $224^2$ |
| Epochs | 300 | 300 | 300 |
| Batch size | 1024 | 4096 | 4096 |
| Optimizer | AdamW | AdamW | AdamW |
| AdamW ($\beta_1, \beta_2$) | 0.9,0.999 | 0.9,0.999 | 0.9,0.999 |
| Learning rate | 0.001 | 0.004 | 0.004 |
| Learning rate decay | Cosine | Cosine | Cosine |
| Weight decay | 0.05 | 0.05 | 0.04/0.05 |
| Warmup epochs | 20 | 20 | 20 |
| Label smoothing $\varepsilon$ | 0.1 | 0.1 | 0.1 |
| Stochastic Depth | ✓ | ✓ | ✓ |
| Rand Augment | 9/0.5 | 9/0.5 | 9/0.5 |
| Repeated Augment | ✓ | ✗ | ✗ |
| Mixup $\alpha$ | 0.8 | 0.8 | 0.1/0.8 |
| CutMix $\alpha$ | 1.0 | 1.0 | 1.0 |
| Erasing prob. | 0.25 | 0.25 | ✗/0.25 |
| ColorJitter | ✗ | ✗ | ✗ |
| Gradient Clipping | ✓ | ✗ | ✗ |
| EMA decay | ✓ | ✓ | ✓ |
| Test crop ratio | 0.875 | 0.875 | 0.90 |

using only a quarter of the parameters of ResNet-34 [71]. When compared to naive detectors with around 20M parameters, the modern networks (Swin-T [59] and ConvNeXt-T [56]) consistently improve the classical CNNs (ResNet-50 [71], Xception [34], and EfficientNet-B4 [35]), e.g., ConvNeXt-T outperforms ResNet-50 and EfficientNet-B4 by 10.29% and 0.77% AUC on FF-c23, which might attribute to the Metaformer macro design [74] and more parameters. Meanwhile, our proposed MkfaNet-S significantly improves both classical CNNs and modern networks with efficient usage of parameters, e.g., MkfaNet-S yields 94.85% AUC and around 0.25∼8.0% performance gains in average comparing to previous backbones. When employing larger backbone encoders with manually-designed features, FFD [62] with MkfaNet-S significantly outperforms frequency detectors (F3Net [19], SPSL [60], and SRM [18]) and spatial detectors (FWA [61], X-ray [28], and CORE [63]) with Xception and HRNet [75] backbones in the similar parameter scale, while even yielding better results than UCF [64] with a larger Xception backbone.

**Cross-domain evaluations.** Then, we evaluate detectors on different datasets without further fine-tuning, which reflects the generalization and robustness of the compared detectors. As shown in Table 4 (right columns), all models suffer performance decreases because of the challenging domain gap. Surprisingly, our proposed MkfaNet variants achieve

the best average results and form greater performance gains over existing methods, e.g., Naive detector with MkfaNet-S outperforms Xception and ConvNeXt-T by 1.43% and 1.13% average AUC, indicating that MkfaNet might learn more common and robust features. We verify this hypothesis in Sect 4.3 with visualizations.

**Further analysis.** The confusion matrix in Table 6 shows that the model misclassified 459 fake images as real (FN) and 181 real images as fake (FP). As observed in Fig 4, false positives are often caused by compression artifacts, occlusions, and complex backgrounds, while false negatives arise from high-quality Deepfakes with minimal detectable artifacts or low-frequency manipulations that retain high-frequency details. These cases highlight the inherent limitations of relying solely on a novel backbone for Deepfake detection.

Meanwhile, the true positive (TP) and true negative (TN) cases demonstrate the model's strong capability in correctly identifying both forged and authentic faces, even under challenging conditions such as lighting variation and facial expression changes. These correctly classified examples highlight the robustness of the proposed MkfaNet in detecting subtle forgery cues and preserving real content recognition.

To further improve performance, multi-task learning could be explored, incorporating auxiliary tasks such as forgery localization, frequency domain analysis, or uncertainty estimation to provide additional supervision. Additionally, integrating temporal consistency analysis for video-based detection or using adaptive feature fusion strategies could enhance robustness against challenging samples. These directions offer promising pathways for advancing Deepfake detection beyond backbone design alone.

### 4.3 Ablation and analysis

**Ablation studies of network modules.** We first ablate the designed modules in MkfaNet with a simplified experimental setting, i.e., training and evaluation on FF-c23 without using ImageNet-1K pre-trained weights. We take ConvNeXt-T [56] as the baseline for MkfaNet, which outperforms the classical bottleneck in ResNet-50 [71] in Table 7. As for the proposed **Multi-kernel Aggregator** (MKA) block, using the **Gating Branch** in Eq 2 can yield similar performances as ConvNeXt-T with around 10M fewer parameters and using **Multi-DWConv**$7 \times 7$ with dilated ratios in (1,2,3) aggregates contextualized patterns and improves the performances. As for the **Multi-Frequency Aggregator** (MFA) block, adding a Squeeze-and-excitation (SE) module [76] to **DWConv**$3 \times 3$ **+ FFN** is equivalent to the EfficientNet block [35], which requires numerous parameters for performance gains. Our proposed **MF** module in Eq 4 brings better AUC than the SE module while using fewer parameters.

**Visualization analysis.** We then evaluate the learned features of MkfaNet-S by two visualizations. As shown in Fig 5, the representations of various detectors are visualized by t-SNE [72] on FF++ (c23) dataset with 5000 randomly selected samples following DeepfakeBench, where four forgery types (Deepfakes, Face2Face, FaceSwap, and NeuralTextures) in FF++ are considered. The representations of real and fake samples are more separable in MkfaNet-S than in previous works, while four different forgeries are also discriminative by MkfaNer-S. It indicates that MkfaNet can capture common features rather than over-fitting the training dataset. Meanwhile, we further investigate the spatial features learned by MkfaNet with Grad-CAM [73] visualization in Fig 6. We consider cross-domain evaluation samples from FFDI-2024, the latest Global Multimedia Deepfake Detection competition at kaggle, and compare with various backbone architectures. Fig 6 shows that MkfaNet-S precisely and consistently locates organs to determine fake or real faces, while other

**Table 6**. **Confusion matrix for the CelebDF-v1 dataset.** This table presents the classification results of our model on the CelebDF-v1 test set, which contains 1,203 real samples and 1,933 fake samples. The confusion matrix includes four key values: True Positive (TP), False Negative (FN), False Positive (FP), and True Negative (TN).

| Actual/Predicted | Real (Positive) | Fake (Negative) |
|---|---|---|
| **Real (P)** | 1022 (TP) | 181 (FN) |
| **Fake (N)** | 459 (FP) | 1474 (TN) |

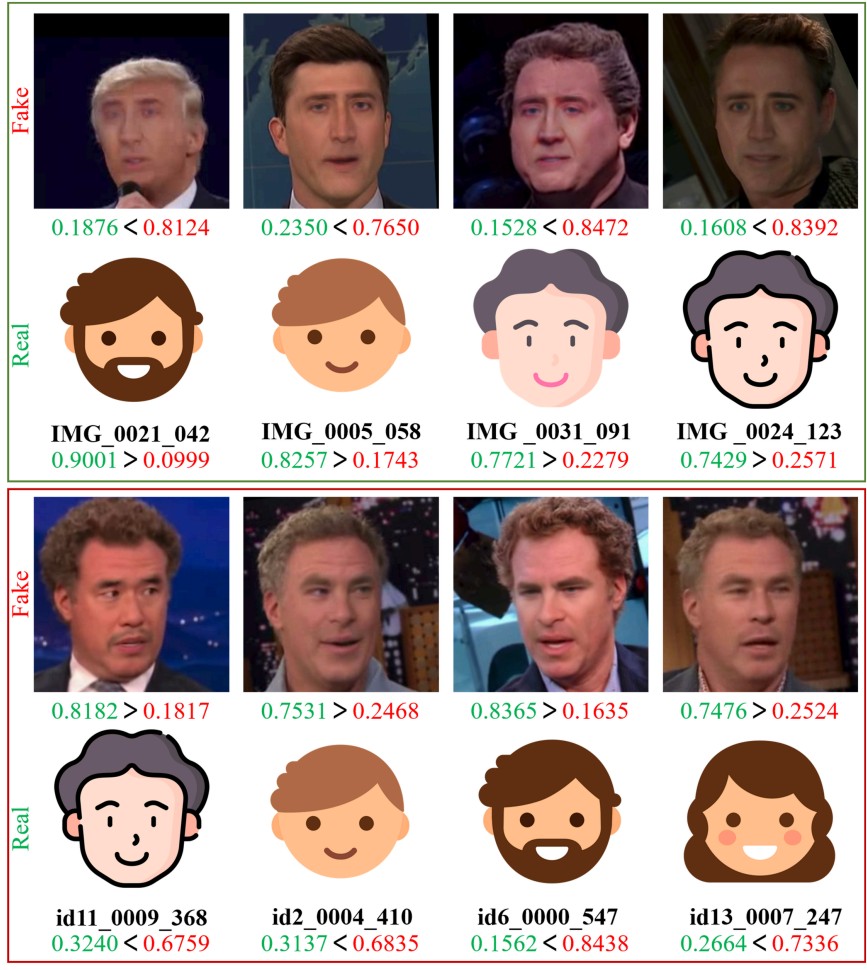

**Fig 4**. **Visualization of true positives, true negatives, false positives, and false negatives in deepfake detection.** This figure illustrates four types of prediction results made by MkfaNet on the CelebDF-v1 dataset. The first row presents true positives (TP), where fake images are correctly classified with high confidence. The second row shows true negatives (TN), where real images are accurately recognized as real. The third row presents false positives (FP), where fake images are mistakenly classified as real. The fourth row shows false negatives (FN), where real images are misclassified as fake due to factors such as compression artifacts, facial expressions, or lighting conditions. The softmax output probabilities indicate the model's prediction confidence. To protect privacy, the original facial images were anonymized by replacing them with icon representations. Corresponding image identifiers from the CelebDF-v1 and UADFV datasets are shown.

backbones sometimes extract irrelevant regions, which might deteriorate the generalization and robustness of forgery detection.

## 5 Conclusion

In this paper, we introduce MkfaNet, a novel backbone network specifically designed for face forgery detection. It combines two core modules, the Multi-Kernel Aggregator (MKA) and the Multi-Frequency Aggregator (MFA), which effectively enhance the ability to distinguish between real and forged facial features. The MKA module targets spatial context by adaptively selecting organ-specific features extracted through multiple convolutions to simulate the subtle facial differences between real and fake faces. The MFA module focuses on frequency components, processing different frequency

**Table 7**. **Ablation of designed modules on FF-c23.** The module without "+" denotes the baseline modules, while those with "+" are added to the baseline (using gray backgrounds). c1, c2, and c3 represent the number of channels assigned to the MKA module's branches with dilation rates of 1, 2, and 3, respectively.

| Block | Module | FF-c23 (AUC) | # Param. (M) |
|---|---|---|---|
| ResNet | Bottleneck | 0.8437 | 25.6 |
| ConvNeXt | DWConv7 × 7+FFN | 0.8856 | 28.5 |
| MKA | Gating Branch | 0.8819 | 18.5 |
| | +DWConv7×7 | 0.8932 | 18.7 |
| | +Multi-DWConv7 × 7 | **0.9015** | 19.0 |
| MFA | DWConv3 × 3+FFN | 0.9093 | 19.2 |
| | +SE | 0.9150 | 21.5 |
| | +MF | **0.9176** | 19.8 |
| MKA | c1:c2:c3=1:3:4 | **0.9176** | 19.8 |
| | c1:c2:c3=1:0:3 | 0.9162 | 19.8 |
| | c1:c2:c3=0:1:1 | 0.9169 | 19.8 |
| | c1:c2:c3=1:6:9 | 0.9174 | 19.8 |

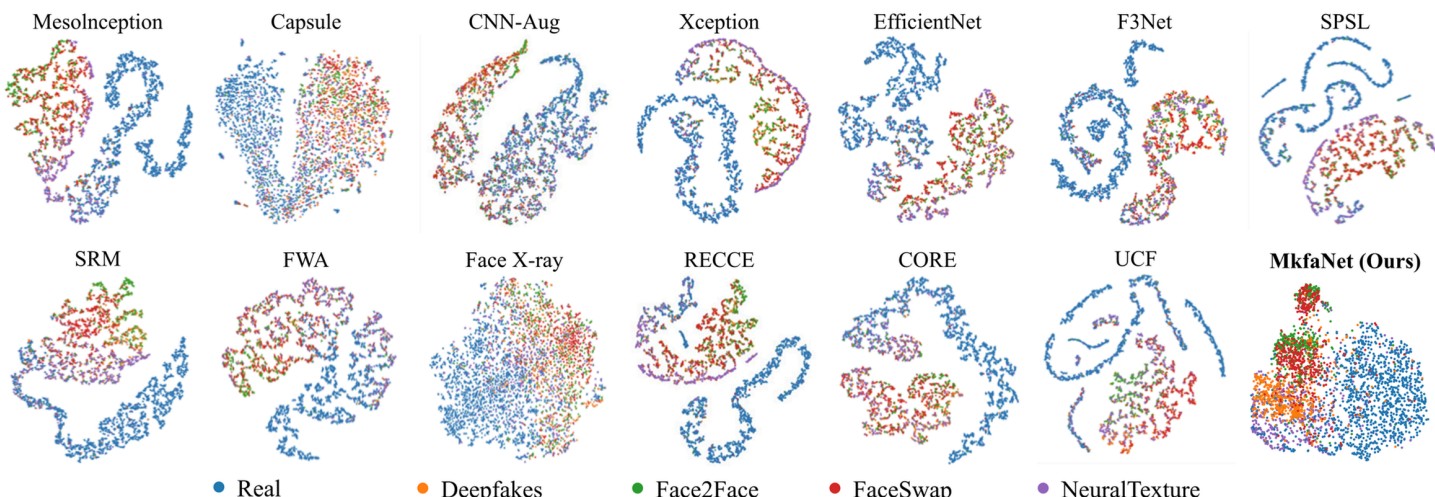

**Fig 5**. **Visualization of latent embedding of detectors with t-SNE [72] on FF++ (c23) according to settings in DeepfakeBench [39].** Based on the naive detector, our MkfaNet-S distinguishes different types of forgery into several clusters, whereas other backbones could not learn the discriminative patterns without additional supervision.

bands by adaptively rebalancing high-frequency and low-frequency features. Comprehensive experiments on seven popular Deepfake detection benchmarks demonstrate that MkfaNet achieves an AUC of 0.9591 in within-domain evaluations and 0.7963 in cross-domain evaluations, outperforming several state-of-the-art methods while maintaining high computational efficiency. This innovative approach not only significantly improves the accuracy of forgery detection but also enhances the model's capability to handle complex facial data, making it a powerful tool for combating advanced forgery techniques in the future.

While MkfaNet demonstrates strong performance across multiple deepfake detection benchmarks, some limitations remain. First, its scalability to extremely large and diverse datasets needs further evaluation, as real-world deepfake videos often exhibit greater variability in quality, compression, and manipulation techniques. Second, its generalization to novel deepfake generation methods is a challenge, as emerging techniques may introduce more sophisticated forgery patterns that require adaptive detection mechanisms.

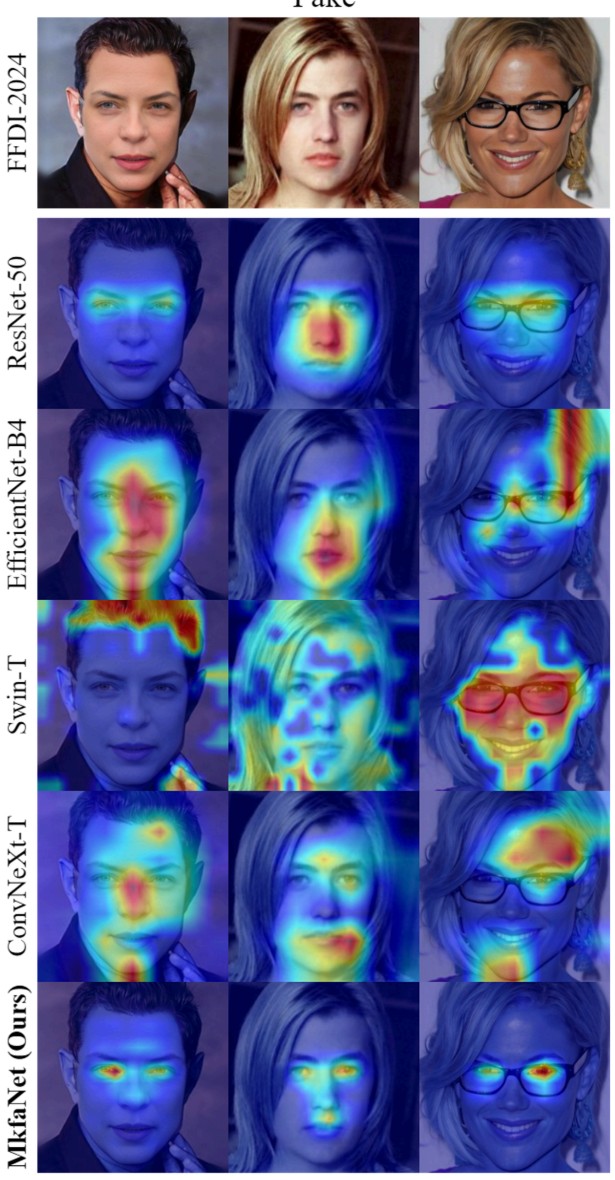

**Fig 6.** Grad-CAM activation maps [73] of fake images in the validation set of FFDI-2024 (collected online) as cross-domain evaluation. Compare the naive detector with different backbones with ours. As for fake images, classical CNNs like ResNet-50 show robust but coarse localization of human faces, while modern architectures like Swin-T can activate some semantic features. Out MkfaNet-S not only exhibits precise localization of discriminative organs but also tells the difference between fake and real faces.

Despite these limitations, our proposed approach significantly improves forgery detection accuracy and enhances the model's ability to handle complex facial data. Future research will focus on adapting MkfaNet to larger datasets and developing more dynamic feature extraction strategies to enhance robustness against evolving forgery techniques. This work provides a solid foundation for future advancements in deepfake detection and contributes to the ongoing fight against digital media manipulation.

## Author contributions

**Data curation:** Shijun Gao.

**Funding acquisition:** Shuai Wang.

**Methodology:** Zifeng Li, Wenzhong Tang, Shijun Gao.

**Supervision:** Shuai Wang.

**Validation:** Zifeng Li.

**Visualization:** Zifeng Li.

**Writing – original draft:** Zifeng Li, Yanyang Wang, Shuai Wang.

**Writing – review & editing:** Zifeng Li, Yanyang Wang, Shuai Wang.

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
