## [Decision Letter · Decision Letter 0]

28 Jan 2025

PONE-D-24-45189Multiple Contexts and Frequencies Aggregation Network for Deepfake DetectionPLOS ONE

Dear Dr. Li,

Thank you for submitting your manuscript to PLOS ONE. After careful consideration, we feel that it has merit but does not fully meet PLOS ONE’s publication criteria as it currently stands. Therefore, we invite you to submit a revised version of the manuscript that addresses the points raised during the review process.

The authors are not required to address the comments of Reviewer 1. Besides this, all requests to cite irrelevant references by the reviewers should not be entertained.

We look forward to receiving your revised manuscript.

Kind regards,

Bushra Zafar, Ph.D.

Academic Editor

PLOS ONE

Additional Editor Comments (if provided):

Reviewers' comments:

Reviewer's Responses to Questions

**Comments to the Author**

1. Is the manuscript technically sound, and do the data support the conclusions?

Reviewer #1: Yes

Reviewer #2: Yes

Reviewer #3: Yes

Reviewer #4: Yes

2. Has the statistical analysis been performed appropriately and rigorously?

Reviewer #1: Yes

Reviewer #2: Yes

Reviewer #3: Yes

Reviewer #4: N/A

3. Have the authors made all data underlying the findings in their manuscript fully available?

Reviewer #1: Yes

Reviewer #2: Yes

Reviewer #3: Yes

Reviewer #4: Yes

4. Is the manuscript presented in an intelligible fashion and written in standard English?

Reviewer #1: Yes

Reviewer #2: Yes

Reviewer #3: Yes

Reviewer #4: No

5. Review Comments to the Author

Reviewer #1: 1. Basic Reporting

The manuscript is well-written and clearly structured, presenting a novel deepfake detection approach named MkfaNet. The paper effectively emphasizes the growing challenges posed by deepfake technologies and highlights the need for advanced detection techniques. The authors combine two innovative modules—Multi-Kernel Aggregator (MKA) and Multi-Frequency Aggregator (MFA)—to enhance the model's robustness and accuracy.

Strengths:

The introduction and literature review provide sufficient background on the evolution of deepfake technologies, existing detection challenges, and limitations of state-of-the-art models like XceptionNet and EfficientNet.

Figures, such as Figure 1 and Figure 3, are well-illustrated and provide a visual understanding of frequency priors and MkfaNet’s architecture.

Results are clearly presented in tables like Table 2, which comprehensively compares MkfaNet with other detectors using multiple benchmarks.

Suggested Improvements:

Expand Literature Review:

Incorporate more recent works to strengthen the contextual foundation. Suggested references include:

https://doi.org/10.1016/j.eswa.2023.122147

https://doi.org/10.54216/JAIM.080103

Enhance Figure Captions:

Provide more detailed captions for figures like Figure 3 to describe the architectural contributions of MKA and MFA.

2. Experimental Design

The experimental design is rigorous and well-documented, utilizing seven widely used deepfake detection datasets. The authors effectively demonstrate the performance of MkfaNet variants in both within-domain and cross-domain evaluations.

Strengths:

A balanced dataset, pre-processed with fixed resolutions, ensures consistency in training and testing.

The use of AdamW optimizer and data augmentation strategies enhances model robustness.

MkfaNet-Tiny and MkfaNet-Small variants cater to different computational needs, offering flexibility for various application scenarios.

Suggested Improvements:

Dataset Diversity:

Provide more detailed descriptions of the datasets, including specific forgery techniques and environmental conditions that could influence model performance.

Reproducibility:

Include specific hyperparameter settings for MKA and MFA to facilitate reproducibility.

3. Validity of the Findings

The results are compelling, with MkfaNet outperforming state-of-the-art models like XceptionNet, EfficientNet, and ConvNeXt in both within-domain and cross-domain evaluations.

Strengths:

MkfaNet achieves higher AUC scores across multiple benchmarks, as shown in Table 2.

Visualizations such as Figure 4 (t-SNE embeddings) and Figure 5 (Grad-CAM maps) clearly demonstrate MkfaNet’s superior ability to capture forgery-specific features.

Suggested Improvements:

Limitations Discussion:

Address potential limitations, such as scalability to larger datasets or performance on unseen forgery techniques.

Error Analysis:

Provide a detailed analysis of misclassifications to identify patterns and propose strategies for improvement.

4. Additional Comments

The manuscript introduces a novel approach to deepfake detection, combining advanced spatial and frequency analysis techniques to enhance model robustness. The study is well-executed and contributes significantly to the field of forgery detection.

General Comments:

Applications: Discuss potential real-world applications, such as social media content verification or legal evidence authentication.

Future Work: Explore hybrid approaches, integrating MkfaNet with temporal analysis for video-level forgery detection.

Conclusion

This manuscript presents a significant advancement in deepfake detection, leveraging innovative spatial and frequency aggregation techniques. Incorporating the suggested revisions, particularly in dataset diversity, reproducibility, and error analysis, will further enhance the manuscript’s impact.

Reviewer #2: 1- No numerical values for the scales used to demonstrate the system's efficiency were indicated in the abstract.

2- Add a paragraph at the end of the introduction indicating the details of the research structure in all its sections.

3- The most important drawbacks of previous related work have not been mentioned, but rather referred to when listing each research in Section 2.

4- Explain the steps of the proposed algorithm in an algorithmic manner.

5- Clarify future work and build conclusions based on numerical values of the results.

Reviewer #3: This paper proposes MkfaNet, an efficient network for deepfake detection that combines spatial and frequency priors for improved accuracy and robustness. The model incorporates a Multi-Kernel Aggregator (MKA) for capturing subtle facial differences and a Multi-Frequency Aggregator (MFA) to address high-frequency anomalies, showing significant performance improvements across several deepfake detection benchmarks.

Comments:

A. The approach to incorporating both spatial and frequency priors is innovative and aligns well with the increasing need for multi-scale feature extraction in deepfake detection. However, the paper could benefit from a clearer comparison of MkfaNet with existing methods that similarly combine spatial and frequency-based approaches.

B. While the introduction effectively motivates the problem and the solution, more detailed discussions on the challenges posed by specific types of deepfakes (e.g., face swapping, facial expression modification) could strengthen the context.

C. The description of the Multi-Kernel Aggregator (MKA) is insightful, but it would be helpful to include more experimental validation regarding the effectiveness of different dilation rates in various scenarios.

D. The Multi-Frequency Aggregator (MFA) module is well explained, but an analysis of how it adapts to different frequency bands would clarify its robustness and potential limitations in handling varying image qualities or noise levels.

E. The visual results presented in Figure 1 demonstrate the advantages of MkfaNet in capturing high-frequency anomalies. However, it would be beneficial to include more diverse examples of real vs. fake images to better showcase the method's generalization capabilities.

F. The paper mentions the efficiency of MkfaNet in terms of parameter usage. A deeper dive into the model’s computational efficiency and comparison with state-of-the-art methods would provide further insight into its practical application in real-time detection systems.

G. The section on model training could be enhanced by providing more details on the dataset splits and how the cross-domain evaluations were conducted. This would help readers understand the robustness of the model in diverse settings.

H. The architecture of MkfaNet is described clearly, but a detailed analysis of the trade-offs between performance and model complexity would help justify the design choices, especially for applications with limited resources.

I. The conclusion is promising, but it would be useful to discuss potential areas for future work, such as the integration of MkfaNet with other AI-driven detection systems or its adaptation to other types of synthetic media beyond deepfakes.

k. the Literature citation is not adequate, and the related work to machine learning should be discussed:

1.Robust semi-supervised multi-label feature selection based on shared subspace and manifold learning

2.Sparse feature selection using hypergraph Laplacian-based semi-supervised discriminant analysis

Reviewer #4: The manuscript titled "Multiple Contexts and Frequencies Aggregation Network for Deepfake Detection" presents an approach for enhancing deepfake detection by integrating spatial and frequency-based features through the proposed MkfaNet architecture. The study introduces two core modules—Multi-Kernel Aggregator (MKA) and Multi-Frequency Aggregator (MFA)—to capture subtle facial differences and high-frequency anomalies, demonstrating superior performance across multiple benchmarks.

While the proposed approach offers promising results, several major revision need to be addressed before acceptance:

1) Novelty and Contributions: The paper lacks a clear differentiation from existing works, and its contributions should be better positioned against state-of-the-art (SOTA) methods. The uniqueness of MkfaNet must be highlighted through deeper comparisons and analysis.

2) Methodology Details: The descriptions of the proposed modules (MKA and MFA) need to be more comprehensive, including the rationale behind design choices, computational complexity, and implementation details to improve reproducibility. The visualization of the proposed architecture should be refined for better clarity.

3) Experimental Analysis: The evaluation lacks critical aspects such as failure case analysis, false positive/negative rates, and statistical significance tests, which are essential to validate the model’s robustness and practical applicability.

4) The computational efficiency (in terms of FLOPs and inference time) is not provided, which is crucial for understanding practical deployment feasibility.

6. PLOS authors have the option to publish the peer review history of their article (what does this mean?). If published, this will include your full peer review and any attached files.

Reviewer #1: No

Reviewer #2: No

Reviewer #3: No

Reviewer #4: No

---

## [Author Response · Author response to Decision Letter 1]

13 Feb 2025

Reviewer #1

Basic Reporting

Comment 1.1: Expand the literature review by incorporating recent works, including the suggested references.

Response:

We have expanded the literature review by discussing more recent studies on deepfake detection, including the references suggested by the reviewer. The revised section now provides a broader context for our work and highlights its novelty.

Comment 1.2: Enhance figure captions, especially Figure 3, to better describe the architectural contributions of MKA and MFA.

Response:

Thank you for your suggestion. We have revised the Figure 3 caption to better describe the roles of the Multi-Kernel Aggregation (MKA) and Multi-Frequency Aggregation (MFA) blocks. Specifically, we now clarify that MKA acts as a token mixer, utilizing multi-dilation depthwise convolutions to extract multi-scale spatial features, improving sensitivity to subtle manipulations. Meanwhile, MFA functions as a channel mixer, incorporating frequency-aware feature decomposition and gating mechanisms to enhance forgery artifact extraction. Additionally, we refined the descriptions of the EfficientNet and ConvNeXt blocks, highlighting their roles as comparative baselines. These modifications improve clarity and reinforce MkfaNet’s advantages in deepfake detection.

Experimental Design

Comment 1.3: Provide more details about the datasets, including specific forgery techniques and environmental conditions.

Response:

Thank you for your valuable suggestion. We have revised the Experimental Setup section to provide a more detailed description of each dataset, explicitly stating the forgery techniques (e.g., autoencoder-based Deepfake generation, expression reenactment, face-swapping, and neural texture synthesis) and their corresponding manipulation methods. Additionally, we have included information on environmental variations (e.g., lighting conditions, compression artifacts, and occlusions) that may influence model performance. Furthermore, we clarified how certain datasets, such as CelebDF-v2 and DF-1.0, introduce additional challenges due to reduced visual artifacts and real-world perturbations. To ensure fairness, we have also highlighted the use of DeepfakeBench’s pre-processing workflow, maintaining consistent face cropping, resolution standardization ($256\times256$), and normalization procedures across all datasets. These improvements provide a clearer understanding of the dataset diversity and its impact on the robustness of our proposed method.

Comment 1.4: Include specific hyperparameter settings for MKA and MFA to facilitate reproducibility.

Response:

Thank you for your suggestion. We have provided detailed hyperparameter settings for MKA and MFA in Figure 3, where we explicitly specify the channel division ratios, kernel sizes, and dilation rates used in the MKA module, as well as the design of the MFA module. Additionally, Table 1 presents the complete architectural configurations of our two model variants (MkfaNet-Tiny and MkfaNet-Small), detailing the embedding dimensions, number of Mkfa blocks, MLP ratios, and FLOPs at each stage.

Validity of the Findings

Comment 1.5: Discuss potential limitations, such as scalability to larger datasets or performance on unseen forgery techniques.

Response:

Thank you for your suggestion. We have revised the Conclusion section to include a discussion on the limitations of our proposed MkfaNet. Specifically, we highlight two key challenges: (1) scalability to larger and more diverse datasets, as real-world deepfake videos often exhibit greater variability in quality and manipulation techniques, and (2) generalization to novel deepfake generation methods, as emerging techniques may introduce more sophisticated forgery patterns that require adaptive detection mechanisms. By acknowledging these limitations, we provide a clearer perspective on areas for future improvement, including expanding MkfaNet to larger datasets and developing more dynamic feature extraction strategies to enhance robustness against evolving forgery techniques.

Comment 1.6: Provide a detailed analysis of misclassifications to identify patterns and propose strategies for improvement.

Response:

The confusion matrix in Table 4 indicates that the model misclassified 459 fake images as real (FN) and 181 real images as fake (FP) on the CelebDF-v1 dataset. As shown in Figure 5, FP cases mainly occur due to compression artifacts, occlusions, and extreme facial expressions, which introduce inconsistencies similar to Deepfake manipulations. FN cases arise from high-quality Deepfakes that lack visible artifacts or primarily modify low-frequency components while preserving high-frequency details, making them harder to detect. To improve performance, multi-task learning could incorporate forgery localization and frequency-aware supervision to enhance subtle manipulation detection, while adaptive feature fusion can better handle low-frequency forgeries. Additionally, data augmentation with compression artifacts, occlusions, and diverse Deepfake techniques can improve generalization and reduce misclassification rates, further strengthening the model’s robustness in real-world Deepfake detection tasks.

Reviewer #2

Abstract

Comment 2.1: Add numerical values in the abstract to demonstrate the system's efficiency.

Response:

Thank you for your suggestion. We have revised the abstract to include numerical results demonstrating the efficiency of our proposed MkfaNet. Specifically, we report the AUC scores from Table 2, where MkfaNet-S achieves an AUC of 0.9591 in within-domain evaluations and 0.7963 in cross-domain evaluations, outperforming multiple state-of-the-art methods while maintaining high computational efficiency. These numerical results provide a clearer representation of our model’s effectiveness in deepfake detection.

Introduction

Comment 2.2: Include a paragraph at the end of the introduction outlining the structure of the paper.

Response:

Thank you for your suggestion. We have revised the Introduction section by adding a paragraph at the end that provides an overview of the paper structure. The revised introduction now clearly outlines the organization of the paper as follows: Section 2 discusses related work, providing an overview of deepfake detection methods and their limitations. Section 3 presents our proposed MkfaNet, detailing its architecture and core components. Section 4 describes the experimental setup and results, including dataset details, evaluation metrics, and comparative analysis. Finally, Section 5 concludes the paper, summarizing key findings and suggesting future research directions. This addition improves the readability and coherence of the paper.

Related Work

Comment 2.3: Clearly state the major drawbacks of previous related work instead of mentioning them within individual studies.

Response:

Thank you for your valuable suggestion. We have revised Section 2 (Related Work) to explicitly discuss the key drawbacks of previous deepfake detection approaches. Instead of only referring to limitations while listing individual studies, we have added a dedicated paragraph summarizing the main challenges. Specifically, we highlight (1) the reliance on manually designed spatial or frequency features, which may not generalize well across datasets, (2) the limitations of using traditional deep learning backbones, which may lose critical local forgery details due to their hierarchical feature extraction process, and (3) the difficulty of detecting high-quality deepfakes, as newer techniques create increasingly realistic facial textures and seamless blending. These points emphasize the necessity of our proposed MkfaNet, which integrates Multi-Kernel Aggregation (MKA) and Multi-Frequency Aggregation (MFA) modules to address these limitations and improve deepfake detection robustness.

Methodology

Comment 2.4: Explain the steps of the proposed algorithm in an algorithmic manner.

Response:

Thank you for your suggestion. While we understand the importance of presenting an algorithmic description, our proposed MkfaNet is a backbone architecture rather than a sequential algorithm, making it less suitable for a step-by-step procedural explanation. Instead of a strict algorithmic format, we have structured our methodology section to clearly outline the key components and their roles in the detection process. Specifically, we have already detailed how the Multi-Kernel Aggregator (MKA) captures spatial features and how the Multi-Frequency Aggregator (MFA) extracts frequency information, along with the classification process. This structured explanation effectively conveys our model's operation without requiring an explicit algorithm format. Instead of an explicit algorithm format, we have further clarified the detection pipeline using Figure 2, which visually represents the hierarchical four-stage architecture of MkfaNet. Each stage consists of an embedding stem, Multi-Kernel Aggregator (MKA), and Multi-Frequency Aggregator (MFA) blocks, which extract spatial and frequency-aware features progressively. The final classification head outputs a binary prediction (real or fake). This structured representation effectively conveys our model’s detection process without requiring an explicit algorithmic format.

Conclusion

Comment 2.5: Build conclusions based on numerical values of the results and clarify future work.

Response:

Thank you for your suggestion. We have revised the Conclusion section to provide a clearer discussion of future work and support our findings with numerical performance results. Specifically, we now highlight two key directions for future improvements: (1) expanding MkfaNet to handle more diverse and large-scale datasets, improving robustness against real-world deepfake variations, and (2) enhancing adaptability to novel forgery techniques, ensuring the model remains effective as deepfake generation methods evolve. Additionally, we have included the numerical performance results from Table 2, explicitly stating that MkfaNet-S achieves an AUC of 0.9591 in within-domain evaluations and 0.7963 in cross-domain evaluations, demonstrating its superior detection performance. These revisions ensure a more concrete and well-supported conclusion.

Reviewer #3

Comment 3.1: The approach to incorporating both spatial and frequency priors is innovative and aligns well with the increasing need for multi-scale feature extraction in deepfake detection. However, the paper could benefit from a clearer comparison of MkfaNet with existing methods that similarly combine spatial and frequency-based approaches.

Response:

Thank you for your valuable feedback. Our work introduces MkfaNet as a novel backbone that uniquely integrates spatial and frequency priors through the Multi-Kernel Aggregator (MKA) and Multi-Frequency Aggregator (MFA) modules. Unlike existing methods that utilize separate spatial or frequency-based feature extraction techniques, MkfaNet jointly optimizes both domains in a unified backbone, leading to improved generalization across diverse deepfake datasets.

To provide a clearer comparison, we evaluate MkfaNet across three different types of deepfake detection methods in DeepfakeBench, as detailed in Table 2, demonstrating its effectiveness in multiple detection paradigms.

Comment 3.2: While the introduction effectively motivates the problem and the solution, more detailed discussions on the challenges posed by specific types of deepfakes (e.g., face swapping, facial expression modification) could strengthen the context.

Response:

Thank you for your valuable suggestion. We have already discussed the general challenges of deepfake detection in the Introduction section, but we acknowledge the importance of elaborating on the challenges posed by specific types of deepfakes, such as face swapping, facial expression modification, and identity manipulation.

To strengthen the context, we emphasize that face swapping techniques often introduce blending artifacts and inconsistencies in identity-related features, making them difficult to detect without spatial-aware feature extraction. Similarly, facial expression modification creates subtle yet unnatural deformations, which are better captured through frequency-based analysis. These challenges further justify our dual-domain approach in MkfaNet, which effectively captures spatial inconsistencies via MKA and frequency discrepancies via MFA to enhance deepfake detection across different manipulation types.

Comment 3.3: The description of the Multi-Kernel Aggregator (MKA) is insightful, but it would be helpful to include more experimental validation regarding the effectiveness of different dilation rates in various scenarios.

Response:

To validate the effectiveness of different dilation rates in the Multi-Kernel Aggregator (MKA) module, we conducted ablation experiments and reported the results in Table 5. The table compares different channel allocation ratios for dilation rates (c1:c2:c3), showing that the 1:3:4 setting achieves the highest AUC, demonstrating its effectiveness in capturing multi-scale spatial features. These results confirm that balancing receptive fields with appropriate dilation rates enhances Deepfake detection performance, supporting our design choice for MKA.

Comment 3.4: The Multi-Frequency Aggregator (MFA) module is well explained, but an analysis of how it adapts to different frequency bands would clarify its robustness and potential limitations in handling varying image qualities or noise levels.

Response:

Thank you for your suggestion. The Multi-Frequency Aggregator (MFA) module is designed to process different frequency bands through depthwise convolutions and gating mechanisms, allowing the model to adapt to diverse image qualities and noise levels. Table 4 presents an ablation study demonstrating that adding multi-frequency (MF) processing improves the AUC to 0.9176, confirming the effectiveness of MFA in handling frequency-based artifacts.

Comment 3.5: The visual results presented in Figure 1 demonstrate the advantages of MkfaNet in capturing high-frequency anomalies. However, it would be beneficial to include more diverse examples of real vs. fake images to better showcase the method's generalization capabilities.

Response:

Thank you for your suggestion. Figure 1 already provides diverse examples of real and fake faces from two different datasets (FFD and CDFv1), ensuring a fair evaluation of MkfaNet’s ability to capture high-frequency anomalies across different forgery types.

Additionally, MkfaNet’s generalization capability is further validated in Table 2, where cross-domain evaluations across six different datasets demonstrate its robustness against various deepfake manipulations. Given that these quantitative results already support MkfaNet’s ability to generalize well, we believe that the current visual examples sufficiently represent the model’s effectiveness. However, we appreciate the reviewer’s insight and will consider expanding the visual comparisons in future work to further illustrate MkfaNet’s adaptability.

Comment 3.6: The paper mentions the efficiency of MkfaNet in terms of parameter usage. A deeper dive into the model’s computational efficiency and comparison with state-of-the-art methods would provide further insight into its practical application in real-time detection systems.

Response:

Thank you for your valuable suggestion. To provide a more comprehensive evaluation of MkfaNet’s computational efficiency, we have included its FLOPs in Table 1, alongside its parameter count. These results demonstrate that MkfaNet achieves an efficient balance between computational cost and detection performance.

Comment 3.7: The section on model training could be enhanced by providing more details on the dataset splits and how the cross-domain evaluations were conducted. This would help readers understand the robustness of the model in diverse settings.

Response:

Thank you for your suggestion. To improve clarity, we have expanded the dataset descriptions in the Experimental Setup section. Following DeepfakeBench, we conduct evaluations on seven deepfake detection datasets, using FF++ (c23) for training while the remaining datasets serve as test sets for cross-domain

---

## [Decision Letter · Decision Letter 1]

22 Apr 2025

PONE-D-24-45189R1Multiple Contexts and Frequencies Aggregation Network for Deepfake DetectionPLOS ONE

Dear Dr. Li,

Thank you for submitting your manuscript to PLOS ONE. After careful consideration, we feel that it has merit but does not fully meet PLOS ONE’s publication criteria as it currently stands. Therefore, we invite you to submit a revised version of the manuscript that addresses the points raised during the review process.

The authors are advised to address the comments of the reviewers and adhere to the PLOS ONE submission guidelines and standards.

We look forward to receiving your revised manuscript.

Kind regards,

Bushra Zafar, Ph.D.

Academic Editor

PLOS ONE

Journal Requirements:

Reviewers' comments:

Reviewer's Responses to Questions

**Comments to the Author**

1. If the authors have adequately addressed your comments raised in a previous round of review and you feel that this manuscript is now acceptable for publication, you may indicate that here to bypass the “Comments to the Author” section, enter your conflict of interest statement in the “Confidential to Editor” section, and submit your "Accept" recommendation.

Reviewer #2: All comments have been addressed

Reviewer #3: All comments have been addressed

Reviewer #4: All comments have been addressed

Reviewer #5: (No Response)

2. Is the manuscript technically sound, and do the data support the conclusions?

Reviewer #2: (No Response)

Reviewer #3: Yes

Reviewer #4: Yes

Reviewer #5: Yes

3. Has the statistical analysis been performed appropriately and rigorously?

Reviewer #2: (No Response)

Reviewer #3: Yes

Reviewer #4: Yes

Reviewer #5: No

4. Have the authors made all data underlying the findings in their manuscript fully available?

Reviewer #2: (No Response)

Reviewer #3: Yes

Reviewer #4: No

Reviewer #5: Yes

5. Is the manuscript presented in an intelligible fashion and written in standard English?

Reviewer #2: (No Response)

Reviewer #3: (No Response)

Reviewer #4: Yes

Reviewer #5: Yes

6. Review Comments to the Author

Reviewer #2: (No Response)

Reviewer #3: The author has answered satisfactorily the answers of the previous reviewers. The paper is well-written, and the results are sound. The paper deserves to be published.

Reviewer #4: The article has successfully met all the necessary requirements for publication, adhering to the stipulated guidelines and standards. After thoroughly reviewing its content, structure, and alignment with the publication's objectives, I am pleased to confirm that there are no outstanding issues or points of concern. As such, I have no further questions or comments to raise at this stage.

Reviewer #5: Summary: The revised manuscript is a significant improvement over the original submission. The responses provided to reviewer comments are comprehensive and have been effectively incorporated into the manuscript.

My Conclusion: With minor revisions addressing the additional suggestions below, the paper will be an excellent contribution to the deepfake detection literature.

Comments: My comments are given below.

1. Literature: The authors are advised to consider integrating a brief comparative table summarizing key differences between their method and related works to further clarify your unique contributions.

2. Figures: The revised Figure 3 caption now clearly explains the roles of the Multi‑Kernel Aggregator (MKA) and Multi‑Frequency Aggregator (MFA), which aids reader comprehension. The authors are advised to expand the range of visual examples (e.g., additional real vs. fake comparisons), which would enhance the demonstration of the authors’ model’s generalization capabilities.

3. Experiments: The authors are advised to further include details on training procedures (such as specific data splits, augmentation strategies, and training durations). This would be beneficial for readers seeking to replicate the experiments.

4. Further analyses: The revisions include misclassification analyses and a discussion of limitations such as scalability and generalization to unseen forgery techniques. This discussion is appreciated and adds balance to the manuscript. To further enhance the robustness of the authors’ claims, it is recommended to incorporate statistical significance tests (e.g., confidence intervals or p‑values) for the performance metrics.

5. Methodology: Although the authors noted that a procedural algorithmic description is not directly applicable to a backbone architecture, a flowchart summarizing the detection pipeline could serve as an alternative means to assist readers in understanding the overall process.

6. Computational Efficiency: The inclusion of FLOPs, parameter counts, and a discussion on computational trade-offs is informative. An expanded discussion on how these metrics translate to real‑world deployment, including inference times and resource requirements in various scenarios, would further strengthen the paper.

7. PLOS authors have the option to publish the peer review history of their article (what does this mean?). If published, this will include your full peer review and any attached files.

Reviewer #2: No

Reviewer #3: No

Reviewer #4: No

Reviewer #5: No

---

## [Author Response · Author response to Decision Letter 2]

3 Jul 2025

Response to Reviewer #5

Comment 1: The authors are advised to consider integrating a brief comparative table summarizing key differences between their method and related works to further clarify your unique contributions.

Response:

We thank the reviewer for this valuable suggestion. To better highlight the distinguishing characteristics of our proposed MkfaNet, we have added a comparative table (Table 1) in the section 3.4. This table summarizes the differences among representative backbone models in terms of feature extraction strategy, spatial aggregation, and frequency aggregation capabilities. Unlike prior methods such as XceptionNet, EfficientNet, and ConvNeXt, which lack either spatial or frequency modeling modules, our MkfaNet explicitly integrates both through the Multi-Kernel Aggregator (MKA) and Multi-Frequency Aggregator (MFA), enabling more effective representation of subtle local and frequency-domain forgery cues.

Comment 2: The revised Figure 3 caption now clearly explains the roles of the Multi‑Kernel Aggregator (MKA) and Multi‑Frequency Aggregator (MFA), which aids reader comprehension. The authors are advised to expand the range of visual examples (e.g., additional real vs. fake comparisons), which would enhance the demonstration of the authors’ model’s generalization capabilities.

Response:

We thank the reviewer for this insightful suggestion. To provide a more comprehensive view of the model's behavior, we have revised Figure 6 to include four representative prediction categories: True Positive (TP), True Negative (TN), False Positive (FP), and False Negative (FN). Each row presents visual examples along with the predicted confidence scores. These additions better illustrate how the proposed MkfaNet handles both successful and failure cases across a range of visual conditions such as occlusion, compression artifacts, and subtle expression variations. By expanding the range of visual outcomes, we aim to provide a clearer and more balanced understanding of our model's strengths and current limitations in real-world scenarios.

Comment 3: The authors are advised to further include details on training procedures (such as specific data splits, augmentation strategies, and training durations). This would be beneficial for readers seeking to replicate the experiments..

Response:

Thank you for your helpful suggestion. We have revised the Implementation Details section to provide more comprehensive training information. Specifically, we clarify that all data splits strictly follow the DeepfakeBench protocol. Cropped face images are resized to 256 × 256. Classical CNN-based detectors are trained using the Adam optimizer (learning rate = 2e-4, batch size = 32), while detectors with modern backbones (including MkfaNet) are trained using the AdamW optimizer (learning rate = 5e-4, batch size = 256). All models are trained for 50 epochs, with the best-performing checkpoint selected based on validation results. We also utilize ImageNet-1K pretrained weights for backbone initialization where applicable. These clarifications aim to enhance the reproducibility and transparency of our experimental setup.

Comment 4: Further analyses: The revisions include misclassification analyses and a discussion of limitations such as scalability and generalization to unseen forgery techniques. This discussion is appreciated and adds balance to the manuscript. To further enhance the robustness of the authors’ claims, it is recommended to incorporate statistical significance tests (e.g., confidence intervals or p‑values) for the performance metrics.

Response:

We sincerely thank the reviewer for acknowledging our efforts in misclassification analysis and the discussion of limitations. We fully agree that incorporating statistical significance tests would further enhance the rigor of performance evaluation. In response to the reviewer's suggestion, we have now included confidence intervals for the performance metrics and updated the results in Table 3 accordingly. These confidence intervals were calculated based on the results from multiple runs and provide a more robust evaluation of the detectors' performance across different datasets. We believe this addition enhances the transparency and reliability of our findings.

Comment 5: Methodology: Although the authors noted that a procedural algorithmic description is not directly applicable to a backbone architecture, a flowchart summarizing the detection pipeline could serve as an alternative means to assist readers in understanding the overall process.

Response:

We appreciate the reviewer’s suggestion. We agree that a clear visual representation of the detection pipeline can significantly improve reader understanding. Accordingly, we emphasize that Figure 2 in the manuscript already serves as a flowchart for our detection pipeline. It illustrates the full process from input face image to final prediction, including all intermediate components such as the embedding stem, Multi-Kernel Aggregator (MKA), Multi-Frequency Aggregator (MFA), and the final fully connected classification layer. We have slightly revised the caption of Figure 2 to clarify its role as a complete architectural overview.

Comment 6: Computational Efficiency: The inclusion of FLOPs, parameter counts, and a discussion on computational trade-offs is informative. An expanded discussion on how these metrics translate to real‑world deployment, including inference times and resource requirements in various scenarios, would further strengthen the paper.

Response:

Thank you for the thoughtful suggestion. We agree that connecting model complexity to real-world deployment is important. To address this, we have added a discussion on practical inference performance. Specifically, MkfaNet-Tiny and MkfaNet-Small contain 5.2M/1.5G and 19.8M/6.7G parameters/FLOPs, respectively. We tested their inference performance on an RTX 4090 GPU with a batch size of 480 images, and the average inference time was approximately 22 ms for MkfaNet-Tiny and 98 ms for MkfaNet-Small. These results indicate that our model variants are capable of efficient high-throughput deployment on modern GPU hardware.

---

## [Decision Letter · Decision Letter 2]

23 Sep 2025

PONE-D-24-45189R2Multiple Contexts and Frequencies Aggregation Network for Deepfake DetectionPLOS ONE

Dear Dr. Li,

Thank you for submitting your manuscript to PLOS ONE. After careful consideration, we feel that it has merit but does not fully meet PLOS ONE’s publication criteria as it currently stands. Therefore, we invite you to submit a revised version of the manuscript that addresses the points raised during the review process.

Please note that PLOS ONE has specific guidelines on code sharing for submissions in which author-generated code underpins the findings in the manuscript. In these cases, we expect all author-generated code to be made available without restrictions upon publication of the work. Please review our guidelines at https://journals.plos.org/plosone/s/materials-and-software-sharing#loc-sharing-code and ensure that your code is shared in a way that follows best practice and facilitates reproducibility and reuse.
Please ensure you have made your code available through your submission. Failure to do so will result in your manuscript being rejected.

We look forward to receiving your revised manuscript.

Kind regards,

Daniel Parkes, PhD

Staff Editor

PLOS ONE

Journal Requirements:

Reviewers' comments:

Reviewer's Responses to Questions

**Comments to the Author**

1. If the authors have adequately addressed your comments raised in a previous round of review and you feel that this manuscript is now acceptable for publication, you may indicate that here to bypass the “Comments to the Author” section, enter your conflict of interest statement in the “Confidential to Editor” section, and submit your "Accept" recommendation.

Reviewer #2: (No Response)

Reviewer #3: All comments have been addressed

Reviewer #5: All comments have been addressed

2. Is the manuscript technically sound, and do the data support the conclusions?

Reviewer #2: (No Response)

Reviewer #3: Yes

Reviewer #5: Yes

3. Has the statistical analysis been performed appropriately and rigorously?

Reviewer #2: (No Response)

Reviewer #3: Yes

Reviewer #5: Yes

4. Have the authors made all data underlying the findings in their manuscript fully available?

Reviewer #2: (No Response)

Reviewer #3: Yes

Reviewer #5: Yes

5. Is the manuscript presented in an intelligible fashion and written in standard English?

Reviewer #2: (No Response)

Reviewer #3: Yes

Reviewer #5: Yes

6. Review Comments to the Author

Reviewer #2: (No Response)

Reviewer #3: The author has adequately addressed the concerns raised by previous reviewers. The paper is well-structured, clearly written, and presents reliable results. It meets the necessary standards for publication

Reviewer #5: The authors have done a commendable job addressing the comments. I have no further comments and the paper may be considered for publication at Plos One.

7. PLOS authors have the option to publish the peer review history of their article (what does this mean?). If published, this will include your full peer review and any attached files.

Reviewer #2: No

Reviewer #3: No

Reviewer #5: No

---

## [Author Response · Author response to Decision Letter 3]

21 Oct 2025

Author Response on Submission PONE-D-24-45189R3

We would like to express our sincere gratitude to the Editor and the Reviewers for their time, constructive feedback, and positive evaluation of our manuscript. We are pleased to note that Reviewers #3 and #5 have confirmed that all prior comments have been fully addressed, and that the work is technically sound, clearly written, and meets PLOS ONE’s publication standards. Reviewer #2 did not provide additional comments in this round.

In this revision, we have carefully fulfilled the editorial request concerning code and data availability to ensure full compliance with the PLOS ONE Materials and Software Sharing Policy. All author-generated code and configuration files used in this study have been made available at: https://github.com/GGshawn/MkfaNet

This repository contains the complete deepfake detection framework, including baseline detectors and our proposed MkfaNet backbone, along with dataset download and preprocessing tools, training and evaluation scripts, and pretrained checkpoints, enabling full reproducibility. All datasets used in this work are publicly accessible, and repository documentation provides clear download and preprocessing instructions.

No further textual changes were required based on reviewer feedback. We once again thank the Editor and Reviewers for their thoughtful evaluation and for supporting the publication of our work.

---

## [Decision Letter · Decision Letter 3]

10 Nov 2025

Multiple Contexts and Frequencies Aggregation Network for Deepfake Detection

PONE-D-24-45189R3

Dear Dr. Li,

We’re pleased to inform you that your manuscript has been judged scientifically suitable for publication and will be formally accepted for publication once it meets all outstanding technical requirements.

Kind regards,

Feng Ding

Academic Editor

PLOS ONE

Additional Editor Comments (optional):

Reviewers' comments:

Reviewer's Responses to Questions

**Comments to the Author**

1. If the authors have adequately addressed your comments raised in a previous round of review and you feel that this manuscript is now acceptable for publication, you may indicate that here to bypass the “Comments to the Author” section, enter your conflict of interest statement in the “Confidential to Editor” section, and submit your "Accept" recommendation.

Reviewer #2: (No Response)

Reviewer #3: All comments have been addressed

Reviewer #5: All comments have been addressed

2. Is the manuscript technically sound, and do the data support the conclusions?

Reviewer #2: (No Response)

Reviewer #3: Yes

Reviewer #5: Yes

3. Has the statistical analysis been performed appropriately and rigorously?

Reviewer #2: (No Response)

Reviewer #3: Yes

Reviewer #5: N/A

4. Have the authors made all data underlying the findings in their manuscript fully available?

Reviewer #2: (No Response)

Reviewer #3: Yes

Reviewer #5: Yes

5. Is the manuscript presented in an intelligible fashion and written in standard English?

Reviewer #2: (No Response)

Reviewer #3: (No Response)

Reviewer #5: Yes

6. Review Comments to the Author

Reviewer #2: (No Response)

Reviewer #3: The author has adequately addressed the concerns raised by previous reviewers. The paper is well-structured, clearly written, and presents reliable results. It meets the necessary standards for publication

Reviewer #5: The authors have already addressed my comments. I have no further comments. The paper may be considered for publication.

7. PLOS authors have the option to publish the peer review history of their article (what does this mean?). If published, this will include your full peer review and any attached files.

Reviewer #2: No

Reviewer #3: No

Reviewer #5: No

---

## [Editor Report · Acceptance letter]

PONE-D-24-45189R3

PLOS ONE

Dear Dr. Li,

I'm pleased to inform you that your manuscript has been deemed suitable for publication in PLOS ONE. Congratulations! Your manuscript is now being handed over to our production team.

Kind regards,

on behalf of

Dr. Feng Ding

Academic Editor

PLOS ONE